# Use the Online Network If You Can: Towards Fast and Stable Reinforcement Learning

**Ahmed Hendawy**[1,2,*]   **Henrik Metternich**[1]   **Théo Vincent**[1,3]   **Mahdi Kallel**[5]
**Jan Peters**[1,2,3,4]   **Carlo D'Eramo**[5]

[1]Technical University of Darmstadt   [2]Hessian.AI   [3]German Research Center for AI (DFKI)
[4]Robotics Institute Germany (RIG)   [5]University of Würzburg

## Abstract

The use of target networks is a popular approach for estimating value functions in deep Reinforcement Learning (RL). While effective, the target network remains a compromise solution that preserves stability at the cost of slowly moving targets, thus delaying learning. Conversely, using the online network as a bootstrapped target is intuitively appealing, albeit well-known to lead to unstable learning. In this work, we aim to obtain the best out of both worlds by introducing a novel update rule that computes the target using the **MIN**imum estimate between the **T**arget and **O**nline network, giving rise to our method, **MINTO**. Through this simple, yet effective modification, we show that MINTO enables faster and stable value function learning, by mitigating the potential overestimation bias of using the online network for bootstrapping. Notably, MINTO can be seamlessly integrated into a wide range of value-based and actor-critic algorithms with a negligible cost. We evaluate MINTO extensively across diverse benchmarks, spanning online and offline RL, as well as discrete and continuous action spaces. Across all benchmarks, MINTO consistently improves performance, demonstrating its broad applicability and effectiveness.

## 1 Introduction

Reinforcement Learning (RL) has demonstrated exceptional performance and achieved major breakthroughs across a diverse spectrum of decision-making challenges. This success spans domains from mastering complex environments like video games (Mnih et al., 2013; 2015; Hessel et al., 2018) and strategic board games (Silver et al., 2016; 2017), to solving high-dimensional problems in continuous control (Haarnoja et al., 2018a; Schulman et al., 2017). Noteworthy applications include learning complex locomotion skills (Haarnoja et al., 2018b; Rudin et al., 2022) and enabling sophisticated, real-world capabilities such as robotic manipulation (Andrychowicz et al., 2020; Lu et al., 2025).The foundation of this success lies primarily in Deep RL, initiated by the introduction of the Deep Q-Network (DQN) (Mnih et al., 2013), which marked the first successful application of deep neural networks in RL. To make that happen, Mnih et al. (2013) introduce various techniques to mitigate mainly the deadly triad issue (Sutton et al., 1998; Van Hasselt et al., 2018) due to the usage of function approximators, off-policy data, and target bootstrapping. Particularly, Mnih et al. (2013) introduce the concept of *target networks* to mitigate the negative impact of the deadly triad issue (Zhang et al., 2021), where the regression target is computed using a lagged copy of the online network to promote stability during training. This prevents the online network from directly chasing its own rapidly changing estimates, thereby mitigating the problem of *moving targets*. A problem that presents an obstacle towards using fresh estimates from the online network. While the target network has been highly successful in improving stability and convergence, it inherently slows down learning since updates are based on delayed targets. This naturally raises an important question: *how can we accelerate learning by leveraging the most recent online estimates while still preserving the stability of the training process?*

---

*Ahmed Hendawy (`ahmed.hendawy@tu-darmstadt.de`) is the corresponding author.
Code is available at `https://github.com/AhmedMagdyHendawy/MINTO`.

Recent studies have suggested that relying solely on the online network for target bootstrapping can improve performance in certain methods (Bhatt et al., 2024; Kim et al., 2019; Kapturowski et al., 2023). Surprisingly, later findings indicate that reinstating a target network in these same approaches can further enhance results (Palenicek et al., 2026; 2025; Gan et al., 2021). Building on these insights, we propose a complementary perspective: leveraging both the online and target networks jointly to compute the regression target, thereby aiming to combine the benefits of *stability* and *fast* learning. In deep RL, the problem of moving targets is especially evident due to the use of neural networks and the resulting uncontrolled fluctuations in the values of unseen states. This issue becomes even more critical in value-based methods, where overestimation bias drives the online estimates to steadily increase over time. These issues motivate the search for an appropriate selection criterion that can mitigate the impact of overestimation bias when employing the online network for target computation, hence *faster*, yet *stable* learning.

Building on this motivation, we propose **MINTO**, a simple yet effective technique that computes regression targets by taking the **MIN**imum estimated value between the **T**arget and **O**nline networks. By relying on the target network when the online estimate is relatively higher, MINTO reduces overestimation bias, alleviates the moving-target problem, and ensures stable learning. At the same time, by incorporating the online network when its estimate is lower, MINTO leverages fresher information, enabling faster learning. Thanks to its simplicity, MINTO can be seamlessly integrated into a wide range of off-policy methods, including both value-based and actor–critic algorithms, across online and offline RL settings. In this work, we present an extensive empirical evaluation showcasing the benefits of our approach. In addition, we benchmark MINTO against related baselines that exploit online estimates, whether for similar objectives or for different purposes. Our contributions can be summarized as follows: we advocate for a principled combination of online and target networks when computing bootstrapped targets, enabling faster learning while preserving stability. We introduce MINTO, a technique that is simple in design yet effective in practice, that computes regression targets as the minimum between online and target estimates, thereby mitigating the moving-target problem and reducing overestimation bias. We further demonstrate MINTO's broad applicability and effectiveness by integrating it into both value-based and actor–critic methods, across online and offline RL settings. Extensive empirical results show that MINTO consistently outperforms conventional target-network designs and related baselines.

## 2 RELATED WORK

Many works focus on developing simpler RL algorithms that are closer to the original Q-learning algorithm to benefit from up-to-date Bellman updates. A reasonable approach is to use additional resources or privileged information to remove the target network. For example, Gallici et al. (2025) demonstrated that cleverly using parallel environments mitigates the need for a target network. However, real-world applications are often limited to a single process. Interestingly, Lindström et al. (2025) show that constructing the regression target from the online network alone is stable after a pre-training phase using expert data. While this study gives hope that a target-free algorithm is feasible, it still relies on additional resources that are not available in the general case. Shao et al. (2022) introduce a cross-entropy method to refine the actions suggested by the policy to construct the bootstrapped estimate. While promising, this technique is limited to the actor-critic setting, and requires additional resources to optimize the landscape defined by the $Q$-function. In this work, we do not consider having access to additional resources or privileged information as we are interested in building a general-purpose algorithm for off-policy learning.

Another approach is to rely on a single estimator, thereby reducing the number of parameters used during training. For example, Kim et al. (2019) replace the maximum operator with the MellowMax operator to construct a target-free algorithm. However, a following work (Gan et al., 2021) suggests that reintroducing the target network is beneficial, indicating that the improvements made by the first approach come from the different nature of the update instead of up-to-date bootstrapped estimates. This makes MellowMax mostly orthogonal to our approach. Kapturowski et al. (2023) explore incorporating recent online estimates through a trust region around the target-network values, yet their method is a comprehensive algorithm with multiple interacting components contributing to performance. Some works intervene in the architecture of the function approximator to stabilize the training dynamics when using only one estimator. For example, Li & Pathak (2021) design neural networks processing the state given as input in the Fourier space. Nonetheless, their analysis shows that the approach struggles to handle high-dimensional input spaces. More recently,

Bhatt et al. (2024) introduce CrossQ, an actor-critic algorithm that uses batch normalization (Ioffe & Szegedy, 2015) to account for the distribution match between the state-action pair and the next state-next action pair. A follow-up work demonstrates that this idea can work better with a target network (Palenicek et al., 2025), moving away from the objective of constructing the regression target from up-to-date estimates, and making this method orthogonal to our approach. In the following, we also choose to rely on a target network since Vincent et al. (2026) show that target-free methods still underperform compared to target-based methods. Closer to our approach are the hybrid methods where the regression target is built from the online network, but an old copy of the online network is used to regularize the update. Zhu & Rigotti (2021) introduce Self-correcting $Q$-learning (ScQL), which evaluates the $Q$-estimate of the next state at the action that maximizes a combination of the target and online network. Piché et al. (2023) regularize the online network predictions with the prediction given by the target network. In Section 5.1, we compare MINTO to those methods.

Building the regression target from the online network accelerates the overestimation bias, which degrades performance. The overestimation bias is a long-standing problem in off-policy RL (Hasselt, 2010), which arises from the interaction between the maximum operator and the stochastic nature of the bootstrap estimate. A first attempt to combat this issue is to learn two independent estimates of the $Q$-function and evaluate one $Q$-estimate on the best action suggested by the other estimator to build the regression target (Hasselt, 2010). While this idea is scalable to deep settings (Van Hasselt et al., 2016), it leads to underestimation, which hinders exploration. The Clipped Double Q-Learning (CDQ) trick, proposed by Fujimoto et al. (2018), is an effective technique for alleviating overestimation bias by taking the minimum between two critic networks. CDQ remains widely used in modern off-policy actor–critic algorithms (Haarnoja et al., 2018b; Lee et al., 2024; 2025; Palenicek et al., 2025) and has inspired several related approaches. For instance, Maxmin Q-learning (Lan et al., 2020) reduces overestimation bias by taking the minimum across several estimators before applying the Bellman maximization operator, but at the cost of training multiple networks. In contrast, our method applies the minimum operator between the target and online network, incorporating the latest online estimates in a stable manner to achieve faster learning.

## 3 PRELIMINARIES

We define the problem as a Markov Decision Process (MDP) (Puterman, 1990), $< \mathcal{S}, \mathcal{A}, P, r, \mu, \gamma >$, where $\mathcal{S}$ is the state space, $\mathcal{A}$ is the action space, $P : \mathcal{S} \times \mathcal{A} \to \Delta(\mathcal{S})$ is the transition distribution where $P(s'|s, a)$ is the probability of reaching state $s'$ from state $s$ after performing action $a$, $r : \mathcal{S} \times \mathcal{A} \to \Delta(\mathbb{R})$ is the reward distribution, $\mu$ is the initial state distribution, and $\gamma \in [0, 1)$ is a discount factor. A policy $\pi : \mathcal{S} \to \Delta(\mathcal{A})$ maps each state to a distribution over the action space. The policy induces an action-value function $Q^\pi(s, a) = \mathbb{E}_\pi[\sum_{t=0}^\infty \gamma^t r(s_t, a_t)|s_0 = s, a_0 = a]$ that defines the expected discounted cumulative return of executing action $a$ in state $s$ while following the policy $\pi$ thereafter. The goal of the agent is to find the policy $\pi$ that maximizes the expected return starting from some initial state. TD learning methods are a suitable set of solutions to this goal, more precisely Q-Learning (Watkins & Dayan, 1992). Q-Learning is an off-policy algorithm that aims to learn the state-action value function $Q$, known as the $Q$-function, of the optimal policy $\pi^*$, $Q^{\pi^*} = Q^*$, by utilizing the Bellman optimality equation (Bellman, 1957):

$$Q^*(s, a) = \mathbb{E}[r(s, a) + \gamma \max_{a' \in \mathcal{A}} Q^*(s', a')]. \tag{1}$$

Therefore, the optimal policy is defined using this value function by acting greedily at a given state $s$, $\pi^* = \text{argmax}_{a \in \mathcal{A}} Q^*(s, a)$. Q-learning offers a recursive update to approximate the state-action value function $Q^*$, given a transition sample $(s, a, r, s')$ generated by any behavioral policy:

$$Q(s, a) \leftarrow Q(s, a) + \alpha(s, a)[y - Q(s, a)], \tag{2}$$

where $y = r + \gamma \max_{a' \in \mathcal{A}} Q(s', a')$ is the target value which is computed via bootstrapping with the current estimate, and $\alpha(s, a)$ is the step-size. In the target value, the maximum expected value of the next state is approximated by applying maximum operator on a single estimate. This introduces a maximization bias and causes Q-learning to overestimate the state-action values (Hasselt, 2010). When dealing with high dimensional state and action spaces, the tabular form of the value functions is not suitable and function approximators are needed. Particularly, in Deep RL, the state-action value function is modeled by a neural network $Q_\theta(s, a)$ with some learnable parameters $\theta$. In Mnih et al. (2013), DQN was introduced as the first successful application of neural networks in RL.

Mnih et al. (2013) introduce a series of algorithmic components, most notably the introduction of the *target network*. Target networks $Q_{\bar{\theta}}$ allow a *stable* learning of value functions by computing the target value using an older copy of the *online network* $Q_\theta$, $y = r + \gamma \max_{a' \in \mathcal{A}} Q_{\bar{\theta}}(s, a)$. This is done by periodically updating the target parameters $\bar{\theta}$ to the online parameters $\theta$ every $T$ steps. Hence preventing the chase of a moving target due to learning $Q_\theta$ from its own value that eventually results in unstable learning. Despite the success, this results in a *slow* learning of the value function as well as the policy due to relying on out-dated estimates. This raises a question: *can we find a practical Bellman update rule that results in a stable and fast learning?*

## 4 METHODOLOGY

### 4.1 MINTO

The aim of this paper is to identify a suitable criterion for incorporating recent online estimates without sacrificing the stability traditionally provided by the target network. This naturally suggests that the online and target networks should work side by side to achieve both fast and stable learning.

To this end, we propose a lightweight and effective technique that leverages online estimates only when they are unlikely to introduce harmful overestimation or rapid fluctuations in the target. In cases where the online estimates are higher than the target one, we instead rely on the target network to ensure stability. Concretely, we achieve this by applying the **MIN**imum operator to the estimated values of the **T**arget and **O**nline networks, giving rise to our method, **MINTO**.

Following our method, the bootstrapped target can be computed as follows:

$$y = r + \gamma \max_{a' \in \mathcal{A}} \min \left( Q_{\bar{\theta}}(s', a'), Q_\theta(s', a') \right). \tag{3}$$

Given the new regression target, we can compute the regression loss, similar in DQN, as follows:

$$\mathcal{L}(\theta) = \tfrac{1}{2} \left( \lceil y \rceil - Q_\theta(s, a) \right)^2, \tag{4}$$

where $\lceil . \rceil$ refers to the *stop gradient* operator, to prevent the backpropagation of gradients to the online network presented in the regression target equation (Eq. 3).

In practice, MINTO can be integrated into DQN and, more generally, into any algorithm that relies on temporal-difference learning (see Appendix B), by modifying only a few lines of code. The method is *lightweight*, requiring only a single additional feedforward pass of the online network on the next state. When implemented in an efficient deep learning framework such as JAX (Bradbury et al., 2018), this overhead is *negligible*.

### 4.2 ANALYZING MINTO

To evaluate the impact of MINTO, we design an empirical study that seeks to answer two central questions: *(Q1) Is the minimum operator an appropriate criterion for combining online and target estimates? (Q2) Can the empirical evidence substantiate the rationale for adopting the minimum operator?* We empirically examine how the minimum operator employed in MINTO stands relative to alternative operators for combining online and target network estimates. Our evaluation is conducted on 15 Atari games and considers the following baselines: **Online Only**, which relies exclusively on the online estimates during training; **Target Only**, equivalent to DQN using solely the target network estimates; **Max**, which selects the larger of the two estimates; **Mean**, which averages them; **Random**, which chooses between the online and target networks with equal probability;

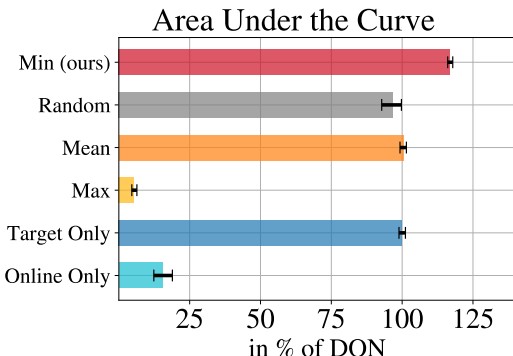

Figure 1: Results of benchmarking the Minimum operator utilized by MINTO against other potential operators on 15 Atari games with the CNN architecture. We report the AUC metric using IQM and the confidence interval computed across 5 seeds. Methods are trained for 50 million frames.

and finally **Min**, the operator at the core of our proposed method, MINTO.

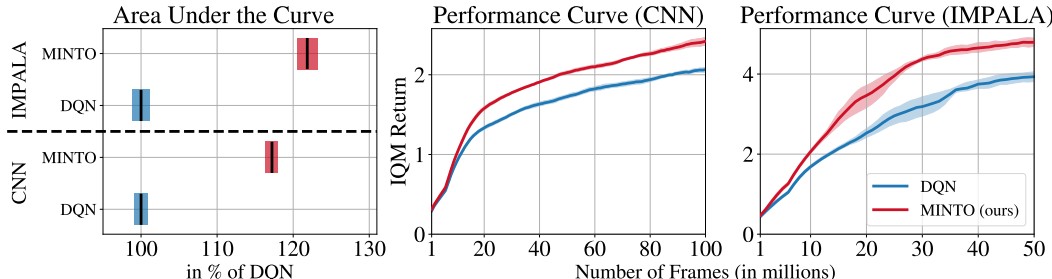

Figure 2: Results of benchmarking MINTO and DQN on 15 Atari games with CNN, and IMPALA with LayerNorm (LN) architectures. All results are reported using IQM and confidence interval on 5 seeds and over all games. **Left**: We report the AUC metric for MINTO and DQN while utilizing both architectures. **Center** and **Right**: We illustrate the performance learning curves of MINTO and DQN under both architecture options.

In Fig. 1, we clearly observe the advantage of the minimum operator over the alternative candidates, thereby addressing *Q1*. As expected, the Online Only function performs poorly, as it relies on rapidly changing bootstrapped targets that lead to unstable training dynamics (see Fig. 33). On the other hand, the Random function fails to match the performance of MINTO, instead converging toward the behavior of Target Only (DQN). The Mean function also aligns closely with Target Only, without offering any additional benefit. Interestingly, the worst results are obtained with the Max operator, which represents the opposite selection criterion to ours. In this case, instability arises from severe overestimation bias, which in turn drives large and uncontrolled increases in target values. Taken together, these findings support our motivation for adopting the minimum operator: it enables the inclusion of recent online estimates in a stable manner by mitigating the overestimation bias they may introduce, thus providing an empirical answer to *Q2*.

### 4.3   CONVERGENCE OF MINTO

While our empirical study highlights the effectiveness of the minimum operator, it is equally important to establish whether its use is theoretically justified in terms of convergence. Since analyzing convergence under function approximation is notoriously difficult, we focus on the tabular setting. Our analysis leverages the Generalized Q-learning framework of Lan et al. (2020), which guarantees convergence for update operators satisfying specific non-expansion properties. Owing to its close relation to Maxmin Q-learning, we show that MINTO can be cast as a special case of this general framework, in the same spirit as how it has been used to analyze other Q-learning variants in their work. This observation leads directly to the following corollary.

**Corollary 1** (Convergence of MINTO). *Let the MINTO operator be defined as $G^{MINTO}(Q_s) = \max_{a \in \mathcal{A}} \min_{j \in \mathcal{T}} Q_{sa}(j)$, where $\mathcal{T}$ is a set of historical time indices. Under the standard stochastic approximation assumptions for the learning rate (Assumption 2 in Lan et al. (2020)), the Q-values generated by the MINTO update rule converge to the optimal action-values, $Q^*$.*

*Proof.* The proof relies on demonstrating that the $G^{MINTO}$ operator satisfies the two conditions on the target operator $G$ (Assumption 1 in Lan et al. (2020)). We provide the detailed verification of these conditions in Appendix A. □

### 5   EXPERIMENTAL RESULTS

We now turn to the empirical evaluation, where we demonstrate MINTO's broad applicability and effectiveness by integrating it into both value-based and actor–critic methods across online and offline RL settings, and show consistent advantages over conventional target-network designs and related baselines.

## 5.1 ONLINE RL AND DISCRETE CONTROL

We evaluate MINTO on a benchmark of 15 Atari games (Bellemare et al., 2013) recommended by Graesser et al. (2022). These games are chosen for their diversity in human-normalized scores achieved by DQN after training. All methods employ the standard **CNN** architecture introduced by Mnih et al. (2015), and our experiments follow the evaluation protocol of Machado et al. (2018) using the official Dopamine hyperparameters (Castro et al., 2018). Additional implementation details and hyperparameters are provided in Appendix C.1.

We begin by evaluating the performance gains of MINTO over DQN (Mnih et al., 2013) in terms of both sample efficiency and asymptotic performance. Our objective is to demonstrate that MINTO not only accelerates learning by incorporating fresher estimates from the online network but also achieves higher final performance. To capture both aspects in a single measure, we report the Area Under the Curve (AUC) metric, alongside learning curves for MINTO and DQN. To further assess the method's effectiveness with more advanced architectures, we extend the comparison to the **IM-PALA** architecture (Espeholt et al., 2018; Castro et al., 2018), evaluating the same metrics. In this setting, we additionally apply **LayerNorm (LN)**, motivated by prior work showing its potential for improving learning stability and performance (Lee et al., 2024; Nauman et al., 2024; Gallici et al., 2025; Vincent et al., 2026).

In Fig. 2, we observe a clear advantage of MINTO over DQN. Specifically, MINTO achieves an improvement of approximately 17% in AUC when using the vanilla CNN architecture, and about 22% when employing IMPALA with LN. The performance curves in Fig. 2 (**center** and **right**) further illustrate MINTO's superior sample efficiency and higher asymptotic performance. These results suggest that incorporating more recent estimates from the online network accelerates learning while also improving final performance.

The idea of leveraging the online network alongside the target network for computing regression targets is not confined to MINTO. Prior work has explored alternative ways of utilizing the online network for different purposes. It is therefore important to evaluate the effectiveness of our selection criterion, the minimum operator, against established methods from the literature. In this study, we compare against three representative baselines: **Double DQN** (Van Hasselt et al., 2016), **Functional Regularization DQN (FR-DQN)** (Piché et al., 2023), and **Self-correcting DQN (ScDQN)** (Zhu & Rigotti, 2021). All methods are implemented with the CNN architecture, and we report aggregate results over all 15 games using the AUC metric. In Fig. 3, we observe that MINTO consistently outperforms all baselines, including methods such as Double DQN and ScDQN

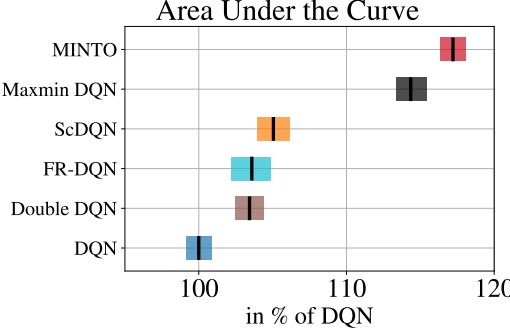

Figure 3: Results of benchmarking MINTO against related baselines on 15 Atari games with the CNN architecture. We report the AUC metric using IQM and the confidence interval computed across 5 seeds.

that are specifically designed to mitigate overestimation. This indicates that combining the online network with the minimum operator provides an especially effective mechanism for addressing this issue. Furthermore, MINTO achieves clear gains over FR-DQN, which relies entirely on the online network for target bootstrapping while regularizing its updates using a fixed network (target network). This highlights the role of our selection criterion, the minimum operator, in regulating the contribution of online estimates. It is also worth noting that methods such as ScDQN and FR-DQN require additional hyperparameters, whereas MINTO introduces none.

Given the similarity of our method to the ensemble-based approach **Maxmin DQN** (Lan et al., 2020), which also employs a minimum operator, it is essential to benchmark MINTO against this prior work. Maxmin DQN applies the minimum operator across multiple estimates produced by an ensemble of target networks, with the goal of reducing the overestimation bias introduced by the maximum operator in the Bellman optimality equation. For this comparison, we also report results in Fig. 3 using the AUC metric. Although Maxmin DQN relies on an ensemble of $Q$-functions ($N = 2$ in our study), MINTO achieves better performance. This demonstrates the advantage of leveraging

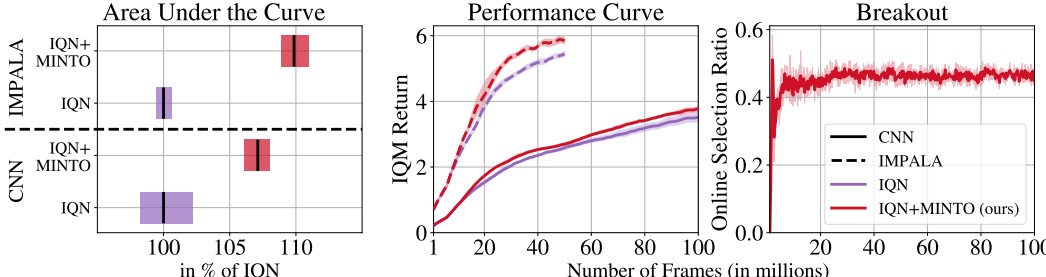

Figure 4: Results of benchmarking MINTO+IQN and IQN on 15 Atari games in the online RL setting with the CNN, and IMPALA with LayerNorm (LN) architectures. All metrics are computed over 5 seeds. **Left**: We report the AUC metric computed by the IQM and its confidence interval. **Center**: We demonstrate the performance learning curves of both methods in terms of IQM return and confidence interval. **Right**: We report the frequency of selecting the online estimate during the course of training on the Breakout game, considering the mean and standard deviation.

up-to-date estimates from the online network, in combination with the minimum operator, to mitigate overestimation bias that can arise from blindly incorporating online estimates. Simultaneously, MINTO avoids the additional memory and compute overhead required by Maxmin DQN.

## 5.2 DISTRIBUTIONAL RL

We also evaluate MINTO within the context of distributional RL (Bellemare et al., 2017), an advanced value-based paradigm that has achieved state-of-the-art performance on the Atari benchmark in the online RL setting (Castro et al., 2018). Specifically, we consider Implicit Quantile Networks (IQN) (Dabney et al., 2018), a distributional RL method that employs quantile regression to approximate the entire return distribution rather than only its expected value. IQN accomplishes this by learning an *implicit quantile function* for the state–action values. For our experiments, we follow the implementation guidelines provided by Castro et al. (2018) when benchmarking IQN on the Atari suite. Comprehensive implementation details and hyperparameter settings are reported in Appendix C.2.

We evaluate the effect of integrating MINTO into IQN by modifying its target computation according to our proposed technique. Appendix B provides further details, including pseudo-code illustrating the changes introduced (see Alg. 2). We benchmark MINTO+IQN against the original IQN on 15 Atari games using the CNN, and the IMPALA with LN architectures. In Fig. 4 (**left**), we report the AUC metric for both methods, which captures improvements in both learning speed and asymptotic performance. According to this metric, MINTO improves IQN by approximately 7% under the CNN architecture and by around 10% under the IMPALA with LN architecture. Fig. 4 (**center**) further presents the learning curves, illustrating consistent gains in sample efficiency and final performance. While the improvement is smaller than in the DQN case (and in later cases we consider), these results clearly demonstrate that MINTO enhances the performance of IQN, establishing it as a general and effective improvement even for state-of-the-art distributional methods.

To verify that the online network is selected frequently during training, we track the online selection ratio and report it for a representative game, Breakout, in Fig. 4 (**right**). The results show that the target network dominates in the very early stages of training, while the use of the online network gradually increases as training progresses. Later in training, the online network is selected approximately 45% of the time. Each point on the curve corresponds to the average online selection ratio between two consecutive target network updates, during which the parameters of the online network increasingly diverge from those of the target network.

## 5.3 OFFLINE RL

The applicability of our method extends beyond online RL. Given its simplicity, MINTO can be readily incorporated into offline RL methods by modifying the Bellman update rule, allowing us to

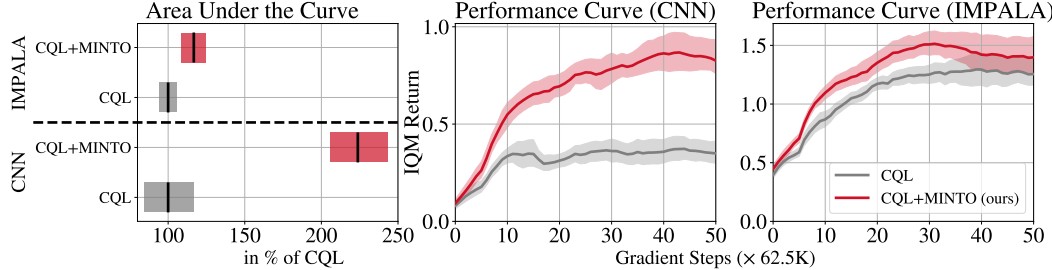

Figure 5: Results of benchmarking CQL and CQL+MINTO on 14 Atari games with CNN, and IMPALA with LayerNorm (LN) architectures. All results are reported using IQM and confidence interval on 5 seeds and over all games. **Left**: We report the AUC metric for both methods while utilizing the two architectures. **Center** and **Right**: We illustrate the performance learning curves of CQL and CQL+MINTO after employing both architecture options.

investigate its potential in this setting. Offline RL aims to learn an optimal policy from a static dataset of previously collected experience, without any further interaction with the environment. A central challenge in this paradigm is the *distributional shift* problem: the learned policy may query the Q-function on state–action pairs absent from the dataset. Overestimation of these out-of-distribution (OOD) actions can then misguide the policy toward suboptimal behaviors.

We consider a popular offline RL algorithm, Conservative Q-Learning (CQL), which regularizes Q-function learning by penalizing OOD values while encouraging in-distribution values. To integrate MINTO into this framework, we simply modify the computation of the regression target using Eq. 3 (see Alg. 3). As in the online setting, we use the hyperparameters from Castro et al. (2018). Additional implementation details and hyperparameters are provided in Appendix C.3.

In Fig. 5, we present the results of evaluating our method, denoted as CQL+MINTO, against the original CQL on 14 Atari games using both the vanilla **CNN** architecture and the **IMPALA** architecture with LN. For the offline setting, we use the datasets provided by Gulcehre et al. (2020). We note that one game, *Tutankham*, is excluded from our evaluation since it is not available in the released dataset.

The results of our experiments demonstrate a clear benefit of applying MINTO to offline RL. MINTO consistently improves the performance of CQL in terms of both sample efficiency and final performance, as reflected in the AUC metric (see Fig. 5 (**left**)) and the learning curves (see Fig. 5 (**center** and **right**)). In particular, MINTO tremendously boosts the performance of CQL with a CNN architecture by roughly 125% in AUC. Although the performance gain is smaller with the IMPALA with LN architecture, MINTO still improves CQL by around 20%, with a clearer sample-efficiency advantage visible in Fig. 5 (**right**). *The improvements are substantial, highlighting the central role of recent online estimates, an aspect overlooked by the original CQL formulation, as well as the effectiveness of the minimum operator in mitigating the overestimation introduced by these estimates.*

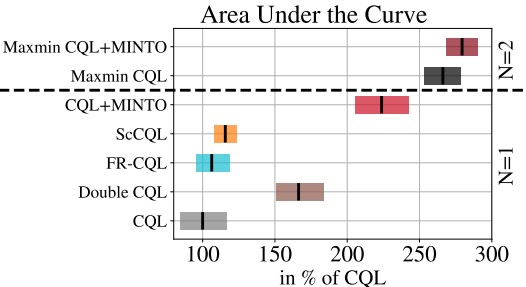

Figure 6: Results of benchmarking MINTO against related baselines on 14 Atari games in the offline RL setting using the CNN architecture. We report the AUC metric using IQM and the confidence interval computed across 5 seeds.

Building on our comparison in Fig. 3, we adapt the online RL baselines to the offline RL setting by using CQL as the underlying RL algorithm. This integration is straightforward due to the close relation between CQL and DQN. As shown in Fig. 6, MINTO consistently outperforms all baselines that rely on a single estimator ($N = 1$), mirroring our observations in the online RL setting. How-

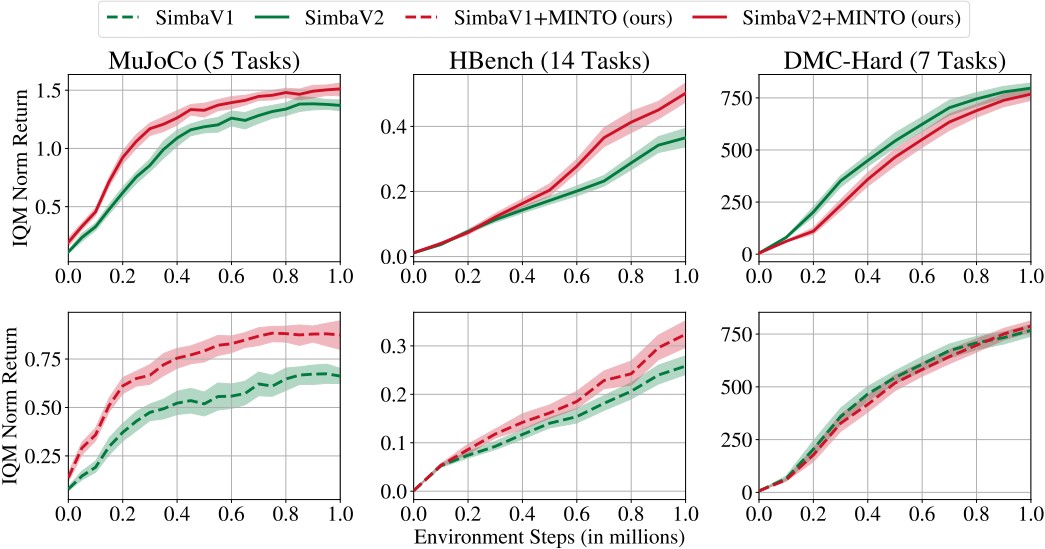

Figure 7: Results of evaluating the impact of MINTO on SimbaV2 (**top**) and SimbaV1 (**bottom**) on three continuous control benchmarks: MuJoCo (**left**), HBench (**center**), and DMC-Hard (**right**). We show the performance curves of all methods on each benchmark while reporting a normalized IQM return and confidence interval computed on 10 seeds.

ever, unlike in the online setting, Maxmin CQL, which employs two estimators ($N = 2$), achieves notably higher performance than MINTO. We attribute this to the conservative nature of Maxmin CQL, whose cautious updates are less detrimental in the offline scenario compared to the online case where exploration plays a major role. This observation motivates us to investigate whether introducing fresh estimates through MINTO into such a strong baseline can yield further gains. By incorporating recent online estimates into the target computation, we find that MINTO indeed enhances the performance of Maxmin CQL, as illustrated in Fig. 6.

## 5.4 ONLINE RL AND CONTINUOUS CONTROL

Similarly, MINTO is not restricted to value-based methods. In off-policy actor–critic algorithms, where a dedicated network models the policy, and the $Q$-function (critic) is trained using bootstrapped targets from a target critic network, our proposed update rule can also be applied. There are multiple ways to incorporate MINTO into actor–critic methods, depending in particular on the number of critic networks employed. We defer a discussion of these design choices to Appendix B.

To examine the effect of incorporating up-to-date estimates in the actor–critic setting, we adopt two recent architectures that are used in combination with **Soft Actor-Critic (SAC)** (Haarnoja et al., 2018a), namely **SimbaV1** (Lee et al., 2024) and **SimbaV2** (Lee et al., 2025). SAC is a widely used actor–critic algorithm that seeks to learn an optimal stochastic policy by encouraging exploration through entropy maximization. In contrast, SimbaV1 and SimbaV2 are recently proposed architectures designed to scale performance with model size by leveraging the concept of simplicity bias, implemented in practice through normalization layers and residual connections. In Alg. 4, we highlight the modifications introduced to SAC in order to integrate MINTO.

We benchmark SimbaV1 and SimbaV2 with and without our proposed method, MINTO. The evaluation is conducted on 26 continuous control tasks, including both manipulation and locomotion, drawn from three different benchmarks: MuJoCo (Todorov et al., 2012), Humanoid Bench (HBench) (Sferrazza et al., 2024), and the DeepMind Control Suite Hard (DMC-Hard) (Tassa et al., 2018). For each actor–critic method, we report the aggregated performance curves on all three benchmarks. Additional details on the experimental setup and hyperparameters are provided in Appendix C.4.

In Fig. 7, we observe a clear improvement in sample efficiency when applying our method on the MuJoCo and HBench benchmarks. In contrast, performance on DMC-Hard is comparable to the

base algorithm (SimbaV1) or slightly lower (as with SimbaV2). Overall, however, MINTO yields a consistent positive impact across all benchmarks and both actor–critic methods, suggesting the value of incorporating recent online estimates when computing regression targets in continuous control and actor–critic settings. Detailed results are in the Appendix D.6 including an additional study on CrossQ+WN (Palenicek et al., 2025).

## 6 CONCLUSION

We introduce MINTO, a simple bootstrapping rule that combines online and target networks by taking their minimum estimate. This design mitigates the overestimation bias that can arise from blindly incorporating online estimates into target computation, thereby alleviating the moving-target problem, while still exploiting recent updates for faster learning. Across online, offline, value-based, and actor–critic methods, MINTO consistently improves sample efficiency and final performance without introducing additional hyperparameters and with only negligible overhead. These results establish MINTO as a practical and effective alternative to conventional target-network designs, pointing toward a promising direction for advancing stable and efficient deep RL. Although MINTO consistently improves performance across diverse settings, there are natural trade-offs to consider. Relying exclusively on the minimum operator may, in some cases, be overly conservative in low-noise environments, leading to slight underestimation. In addition, by dampening optimistic estimates, MINTO may interact with exploration strategies in ways that are not yet fully understood. These observations open promising avenues for future research, such as adaptive operator selection that dynamically balance online and target estimates based on uncertainty or learning dynamics. Beyond these considerations, a separate line of future work lies in testing MINTO in additional challenging scenarios, such as multi-task and multi-agent RL, to assess its scalability and broader applicability.

### ACKNOWLEDGMENTS

The authors would like to thank Daniel Palenicek and Florian Vogt for their valuable support in conducting the study on CrossQ+WN (Palenicek et al., 2025). This work was funded by Hessian.AI through the project 'The Third Wave of Artificial Intelligence – 3AI' by the Ministry for Science and Arts of the state of Hessen. This work was also funded by the Deutsche Forschungsgemeinschaft (DFG, German Research Foundation) under Germany's Excellence Strategy (EXC-3057/1 'Reasonable Artificial Intelligence', Project No. 533677015). Calculations for this research were conducted on the Lichtenberg high-performance computer and the Intelligent Autonomous Systems (IAS) cluster at TU Darmstadt. Furthermore, the authors gratefully acknowledge the scientific support and HPC resources provided by the Erlangen National High Performance Computing Center (NHR@FAU) of the Friedrich-AlexanderUniversität Erlangen-Nürnberg (FAU) under the NHR project b187cb. NHR funding is provided by federal and Bavarian state authorities. NHR@FAU hardware is partially funded by the German Research Foundation (DFG) – 440719683. The authors also gratefully acknowledge the "Julia 2" HPC provided by the Julius-MaximiliansUniversity Würzburg. "Julia 2" was funded as DFG project as "Forschungsgroßgerät nach Art 91bGG" under INST 93/1145-1 FUGG.

### REPRODUCIBILITY STATEMENT

We have taken several measures to ensure the reproducibility of our results. To further promote transparency and replication, we publicly release our code at `https://github.com/AhmedMagdyHendawy/MINTO`. To illustrate the simplicity of our approach, we provide in the appendix both a JAX code snapshot showing how MINTO can be integrated into DQN with minimal changes to the target computation and pseudocode for all algorithms where we integrate our method. Detailed descriptions of the experimental setup, architectures, and hyperparameters are provided in Appendix C. For theoretical results, we clearly state assumptions and include the complete proof of convergence in Appendix A. For empirical evaluation, we report results across multiple random seeds, following standard evaluation protocols (Machado et al., 2018), and provide per-task breakdowns in Appendix D. Offline RL experiments use publicly available datasets (RL Unplugged), and we specify all data processing and training procedures in the appendix. Together, these details should facilitate exact replication and further validation of our findings.

LARGE LANGUAGE MODEL USAGE

A large language model was helpful in polishing writing, improving reading flow, and identifying remaining typos.

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

# APPENDIX

# A    PROOF OF COROLLARY 1

To prove the convergence of MINTO, we show that its target operator, $G^{\text{MINTO}}$, satisfies the two conditions of Assumption 1 from the Generalized Q-learning framework Lan et al. (2020).

**Assumption 1 (Conditions on G)** *Let* $G : \mathbb{R}^{n \times N \times K} \mapsto \mathbb{R}$ *be the target operator where* $Q_s = (Q_{sa}^{ij}) \in \mathbb{R}^{n \times N \times K}$, $a \in \mathcal{A}$, *and* $|\mathcal{A}| = n$, $i \in \{1, \ldots N\}$, $j \in \{0, \ldots, K-1\}$, *and* $s \in \mathcal{S}$ *is an arbitrary state. G must satisfy:*

**A1.1:** If all input action-values are identical, $Q_{sa}^{ij} = Q_{sa}^{kl}, \forall i, k, \forall j, k$, and $\forall a$, then
$G(Q_s) = \max_a Q_{sa}^{ij}$.

**A1.2:** G is a non-expansion w.r.t. the max norm, $|G(Q_s) - G(Q'_s)| \leq \max_{a,i,j} |Q_{sa}^{ij} - Q_{sa}'^{ij}|$.

The action-value function $Q_s$ is represented by a tensor with an ensemble of $N$ action-value functions and for each, $K$ historical time action-values.

The MINTO operator is defined as $G^{\text{MINTO}}(Q_s) = \max_{a \in \mathcal{A}} \left( \min_{j \in \mathcal{T}} Q_{sa}(j) \right)$ for a given state $s$, where $\mathcal{T}$ is a set of historical time indices, and $j$ is the time index with an abuse of notations. This is a special case of the Generalized Q-learning where $N = 1$ and $K = 2$, hence dropping the $i$ index. In the analysis, we consider a general variant of MINTO, where $K > 1$, by considering a set of historical time values using the indices set $\mathcal{T}$ such that $Q_{sa} = \left( Q_{sa}(t-K), \ldots, Q_{sa}(t-1) \right)$ for a given state and action. In practice, $\mathcal{T}$ includes only $(t-1)$ and $(t-K)$, representing the online and target network estimates, respectively.

## A.1    PROOF OF CONDITION A1.1

Assume the input Q-values $Q_{sa}(j)$ are identical for all $j \in \mathcal{T}$ and all $a \in \mathcal{A}$.

$$G^{\text{MINTO}}(Q_s) = \max_{a \in \mathcal{A}} \left( \min_{j \in \mathcal{T}} Q_{sa}(j) \right)$$
$$= \max_{a \in \mathcal{A}} Q_{sa}(j)$$

This is the maximum action-value among the inputs. Thus, A1.1 is satisfied.

## A.2    PROOF OF CONDITION A1.2

Let $Q_s$ and $Q'_s$ be two distinct sets of historical Q-values for a given state $s$, assuming that $N = 1$.

$$|G^{\text{MINTO}}(Q_s) - G^{\text{MINTO}}(Q'_s)| = \left| \max_{a \in \mathcal{A}} \left( \min_{j \in \mathcal{T}} Q_{sa}(j) \right) - \max_{a \in \mathcal{A}} \left( \min_{j \in \mathcal{T}} Q'_{sa}(j) \right) \right| \tag{5}$$

$$\leq \max_{a \in \mathcal{A}} \left| \min_{j \in \mathcal{T}} Q_{sa}(j) - \min_{j \in \mathcal{T}} Q'_{sa}(j) \right| \tag{6}$$

$$\leq \max_{a \in \mathcal{A}} \left( \max_{j \in \mathcal{T}} |Q_{sa}(j) - Q'_{sa}(j)| \right) \tag{7}$$

$$= \max_{a \in \mathcal{A}, j \in \mathcal{T}} |Q_{sa}(j) - Q'_{sa}(j)| \tag{8}$$

The first inequality holds because the max operator is a non-expansion. The second inequality holds because the min operator is also a non-expansion.

The final term is the maximum absolute difference over the subset of Q-values used by the MINTO operator. This maximum is necessarily less than or equal to the maximum over the entire set of all possible Q-values, i.e.:

$$\max_{a \in \mathcal{A}, j \in \mathcal{T}} |Q_{sa}(j) - Q'_{sa}(j)| \leq \max_{a,j} |Q_{sa}(j) - Q'_{sa}(j)|$$

Therefore, $|G^{\text{MINTO}}(Q_s) - G^{\text{MINTO}}(Q'_s)| \leq \max_{a,j} |Q_{sa}(j) - Q'_{sa}(j)|$. Thus, A1.2 is satisfied.

**Conclusion**: Since both conditions of Assumption 1 are satisfied, the convergence of MINTO is guaranteed by Theorem 2 of Lan et al. (2020) under the standard stochastic approximation assumptions for learning rate stated in Assumption 2 of Lan et al. (2020).

## B ALGORITHMIC DETAILS

The pseudocode blocks below present MINTO and its extension to IQN, CQL, and SAC in Alg. 1, Alg. 2, Alg. 3, and Alg. 4 respectively. Changes in the pseudocode of the underlying algorithms are colored red. Note that $\lceil \cdot \rceil$ denotes the stop-gradient operator. For MINTO we want to explicitly stop the gradient, as the target computation relies on both the online as well as the target parameters. In contrast to other methods, like DoubleDQN, for example, the gradient is not stopped by the $\arg\max$ operator, as the online values are used directly.

---

**Algorithm 1** MINTO

---

1: Initialize online and target paramters $\bar{\theta}, \theta$, an empty replay buffer $\mathcal{B}$, and $t_{\text{total}} = 0$.
2: **repeat**
3:      Sample an initial state $s_0$ from $\mu$
4:      **for** $t = 0$ **to** $n_{\text{horizon}}$ **do**
5:          Sample an action $a_t \sim \epsilon$-greedy$(Q_\theta(s_t, \cdot))$
6:          Execute action $a_t$ in environment, observe reward $r_t$ and next state $s_{t+1}$
7:          Store transition $(s_t, a_t, r_t, s_{t+1})$ in $\mathcal{B}$
8:          Sample a batch of $B$ transitions $(s_b, a_b, r_b, s_b')_{b=1}^{B}$ from $\mathcal{B}$
9:          Compute the TD-target $y_b = r_b + \gamma \max_{a'} \min(Q_{\bar{\theta}}(s_b', a'), Q_\theta(s_b', a'))$
10:         **if** $s_b'$ is terminal **then** $y_b \leftarrow r_b$
11:         Compute the loss $\mathcal{L}(\theta) = \frac{1}{2B} \sum_{b=1}^{B} (\lceil y_b \rceil - Q_\theta(s_b, a_b))^2$
12:         Obtain the gradient $\nabla_\theta \mathcal{L}$ and perform an update step
13:         Every $T$ steps update the target network $\bar{\theta} \leftarrow \theta$
14:         **if** $s_{t+1}$ is terminal **break**
15:      **end for**
16:      $t_{\text{total}} \leftarrow t_{\text{total}} + t$
17: **until** $t_{\text{total}} \geq n_{\text{total}}$

---

**Algorithm 2** IQN+MINTO

---

1: Initialize online and target network parameters $\theta, \bar{\theta}$, and replay buffer $\mathcal{B}$, and $t_{\text{total}} = 0$.
2: **repeat**
3:      Sample initial state $s_0 \sim \mu$
4:      **for** $t = 0$ **to** $n_{\text{horizon}}$ **do**
5:          Select action $a_t \sim \epsilon$-greedy$\left( \frac{1}{N} \sum_{i=1}^{N} Z_\theta(s_t, a, \tau_i) \right)$ where $\tau_i \sim U[0,1]$
6:          Execute $a_t$, observe reward $r_t$ and next state $s_{t+1}$
7:          Store transition $(s_t, a_t, r_t, s_{t+1})$ in $\mathcal{B}$
8:          Sample minibatch $(s_b, a_b, r_b, s_b')_{b=1}^{B}$ from $\mathcal{B}$
9:          Sample $\{\tau_i\}_{i=1}^{N}, \{\tau_j'\}_{j=1}^{N'} \sim U[0,1]$
10:         Compute target quantiles:

$$a^* = \arg\max_{a'} \min\left( \frac{1}{N} \sum_{i=1}^{N} Z_{\bar{\theta}}(s_b', a', \tau_i), \frac{1}{N} \sum_{i=1}^{N} Z_\theta(s_b', a', \tau_i) \right)$$

$$y_j = r_b + \gamma \min\left( Z_{\bar{\theta}}(s_b', a^*, \tau_j'), Z_\theta(s_b', a^*, \tau_j') \right)$$

11:         **if** $s_b'$ terminal **then** $y_j \leftarrow r_b$
12:         Compute quantile regression loss:

$$\mathcal{L}(\theta) = \frac{1}{NN'B} \sum_{b=1}^{B} \sum_{i=1}^{N} \sum_{j=1}^{N'} \rho_\kappa^{\tau_i}(\lceil y_j \rceil - Z_\theta(s_b, a_b, \tau_i))$$

13:         Perform gradient step on $\nabla_\theta \mathcal{L}$
14:         Every $T$ steps update target network $\bar{\theta} \leftarrow \theta$
15:         **if** $s_{t+1}$ is terminal **break**
16:      **end for**
17:      $t_{\text{total}} \leftarrow t_{\text{total}} + t$
18: **until** $t_{\text{total}} \geq n_{\text{total}}$

---

---

**Algorithm 3** CQL+MINTO

---

1: Initialize online and target critic parameters $\theta, \bar{\theta}$, an empty replay buffer $\mathcal{B}$, and $t_{\text{total}} = 0$.
2: Load offline dataset $\mathcal{D}$ into $\mathcal{B}$
3: **repeat**
4:     Sample a batch of $B$ transitions $(s_b, a_b, r_b, s'_b)_{b=1}^{B}$ from $\mathcal{B}$
5:     Compute target Q-values for next states:

$$y_b = r_b + \gamma \max_{a'} \min(Q_{\bar{\theta}}(s'_b, a'), Q_{\theta}(s'_b, a'))$$

6:     **if** $s'_b$ is terminal **then** $y_b \leftarrow r_b$
7:     Compute standard TD loss:

$$\mathcal{L}_{\text{TD}}(\theta) = \frac{1}{2B} \sum_{b=1}^{B} \left( \lceil y_b \rceil - Q_{\theta}(s_b, a_b) \right)^2$$

8:     Compute conservative regularizer:

$$\mathcal{L}_{\text{CQL}}(\theta) = \alpha \cdot \frac{1}{B} \sum_{b=1}^{B} \left[ \log \sum_{a} \exp(Q_{\theta}(s_b, a)) - Q_{\theta}(s_b, a_b) \right]$$

9:     Total loss:
$$\mathcal{L}(\theta) = \mathcal{L}_{\text{TD}}(\theta) + \mathcal{L}_{\text{CQL}}(\theta)$$

10:     Update critic parameters: $\theta \leftarrow \theta - \eta \nabla_{\theta} \mathcal{L}(\theta)$
11:     Every $T$ steps update target network: $\bar{\theta} \leftarrow \theta$
12:     $t_{\text{total}} \leftarrow t_{\text{total}} + 1$
13: **until** $t_{\text{total}} \geq n_{\text{steps}}$

---

In Alg. 4, SAC is adapted to use a single Q-function critic, following the approach taken in Simba (Lee et al. (2024; 2025)) on many benchmarks, which also relies on a single critic. This choice motivated our design, as it simplifies computation while remaining effective when combined with the MINTO operator in the target computation. In practice, we also experimented with a double-critic setup (see Fig. 18-21, 24, and 26), where we evaluate the Clipped Double Q-Learning (CDQ) trick with and without MINTO integration of online estimates.

For MINTO integration with CDQ, we considered two variants. In the first variant (as in SimbaV1 and SimbaV2), the online estimates from both critics are aggregated to compute a single shared target value, which is then used to update both networks. In the second variant (as in CrossQ+WN), each critic is updated using only its corresponding online estimate when forming the target. This is a design choice, and a more detailed investigation is left for future work.

In general, the largest performance gains were observed with the single-critic variant, since applying the minimum operator on all online and target networks, in the CDQ case, can lead to overly conservative updates. Nevertheless, the performance gains remain clear after integrating MINTO into the CDQ trick, especially as shown in Fig. 24 and 26 when using the CrossQ+WN (Palenicek et al., 2025) algorithm. The stochastic policy and temperature updates are retained as in standard SAC.

Overall, the MINTO modifications presented in Alg. 1, 2, 3, and 4 are straightforward to implement and can be integrated with minimal changes to the underlying algorithms. This makes MINTO a practical approach for extending existing value-based and actor-critic methods without introducing significant complexity.

---

**Algorithm 4** SAC+MINTO.

---

1: Initialize policy parameters $\phi$, critic's online and target parameters $\theta, \bar{\theta}$, an empty replay buffer $\mathcal{B}$, and $t_{\text{total}} = 0$.
2: **repeat**
3:     Sample an initial state $s_0$ from $\mu$
4:     **for** $t = 0$ **to** $n_{\text{horizon}}$ **do**
5:         Sample action $a_t \sim \pi_\phi(\cdot|s_t)$
6:         Execute $a_t$ in environment, observe reward $r_t$ and next state $s_{t+1}$
7:         Store transition $(s_t, a_t, r_t, s_{t+1})$ in $\mathcal{B}$
8:         Sample a batch of $B$ transitions $(s_b, a_b, r_b, s_b')_{b=1}^B$ from $\mathcal{B}$
9:         Sample $a_b' \sim \pi_\phi(\cdot|s_b')$ and compute $\log \pi_\phi(a_b'|s_b')$
10:       Compute target:

$$y_b = r_b + \gamma \big[ \min \left( Q_{\bar{\theta}}(s_b', a_b'), Q_\theta(s_b', a_b') \right) - \alpha \log \pi_\phi(a_b'|s_b') \big]$$

11:       **if** $s_b'$ is terminal **then** $y_b \leftarrow r_b$
12:       Update critic by minimizing

$$\mathcal{L}(\theta) = \tfrac{1}{2B} \sum_{b=1}^B \left( \lceil y_b \rceil - Q_\theta(s_b, a_b) \right)^2$$

13:       Update policy $\phi$ using gradient of

$$J_\pi(\phi) = \tfrac{1}{B} \sum_{b=1}^B \big( \alpha \log \pi_\phi(a_b|s_b) - Q_\theta(s_b, a_b) \big)$$

14:       Optionally update temperature $\alpha$ by minimizing

$$J(\alpha) = \tfrac{1}{B} \sum_{b=1}^B -\alpha \big( \log \pi_\phi(a_b|s_b) + \mathcal{H}_{\text{target}} \big)$$

15:       Every $T$ steps update target critic: $\bar{\theta} \leftarrow \tau\theta + (1-\tau)\bar{\theta}$
16:       **if** $s_{t+1}$ is terminal **break**
17:     **end for**
18:     $t_{\text{total}} \leftarrow t_{\text{total}} + t$
19: **until** $t_{\text{total}} \geq n_{\text{total}}$

---

# C  IMPLEMENTATION DETAILS

## C.1  ONLINE RL AND DISCRETE CONTROL

For our online reinforcement learning experiments in the discrete-action setting, we use Atari environments as the benchmark domain. The implementations of DoubleDQN, FR-DQN, ScDQN, and MINTO share the same lightweight and reproducible framework for DQN variants written in JAX. The codebase[1] has been released. It is inspired by the code[2] released by Vincent et al. (2025b). All algorithms share the same network architectures and training setups, differing only in the algorithm-specific modifications detailed in Tables 1 and 2. MINTO is implemented by modifying the target computation.

Listing 1: JAX implementation of the MINTO TD target.

```
def compute_minto_target(
    self,
    target_params,
    online_params,
    sample,
):
    q_online_next = self.network.apply(online_params,
        ↪ sample.next_state)
    q_online_next = jax.lax.stop_gradient(q_online_next)
    q_target_next = self.network.apply(target_params,
        ↪ sample.next_state)
    q_next = jnp.max(jnp.minimum(q_online_next, q_target_next))
    return (
        sample.reward
        + (1 - sample.is_terminal) * (self.gamma**self.update_horizon)
            ↪ * q_next
    )
```

| Hyperparameter | —— DQN | —— Maxmin DQN | —— MINTO |
|---|:---:|:---:|:---:|
| Replay Buffer Capacity | | 1,000,000 | |
| Batch Size | | 32 | |
| Update Horizon | | 1 | |
| Discount Factor ($\gamma$) | | 0.99 | |
| Learning Rate | | $6.25 \times 10^{-5}$ | |
| Horizon | | 27,000 | |
| Architecture Type | | CNN | |
| Features | | [32, 64, 64, 512] | |
| Epochs | | 100 | |
| Training Steps per Epoch | | 250,000 | |
| Data to Update | | 4 | |
| Initial Samples | | 20,000 | |
| Epsilon End | | 0.01 | |
| Epsilon Duration | | 250,000 | |
| Target Update Frequency ($T$) | | 8,000 | |
| Ensemble Size ($N$) | - | 2 | - |

Table 1: Hyperparameter settings for DQN, Maxmin DQN, and MINTO on Atari. Most parameters are shared, with algorithm-specific hyperparameters explicitly listed. Identical values are merged for clarity.

---

[1] https://github.com/AhmedMagdyHendawy/MINTO
[2] https://github.com/slimRL/slimDQN

| Hyperparameter | —— DoubleDQN | —— FR-DQN | —— ScDQN | —— MINTO |
|---|---|---|---|---|
| Replay Buffer Capacity | 1,000,000 | | | |
| Batch Size | 32 | | | |
| Update Horizon | 1 | | | |
| Discount Factor ($\gamma$) | 0.99 | | | |
| Learning Rate | $6.25 \times 10^{-5}$ | | | |
| Horizon | 27,000 | | | |
| Architecture Type | CNN | | | |
| Features | [32, 64, 64, 512] | | | |
| Epochs | 100 | | | |
| Training Steps per Epoch | 250,000 | | | |
| Data to Update | 4 | | | |
| Initial Samples | 20,000 | | | |
| Epsilon End | 0.01 | | | |
| Epsilon Duration | 250,000 | | | |
| Target Update Frequency ($T$) | 8,000 | | | |
| Regularization Parameter ($\kappa$) | - | 1.0 | | - |
| Self Correcting Parameter ($\beta$) | - | | 3.0 | - |

Table 2: Hyperparameter settings for the four DQN variants evaluated on Atari, including DoubleDQN, FR-DQN, ScDQN, and MINTO. Most parameters are shared across all algorithms, while the table explicitly lists the algorithm-specific hyperparameters, the regularization parameter $\kappa$ for FR-DQN and the self-correcting parameter $\beta$ for ScDQN. Identical values are merged for clarity.

## C.2 DISTRIBUTIONAL RL

For the distributional reinforcement learning experiments, we use Implicit Quantile Networks (IQN). The implementation builds directly on the same JAX codebase as the DQN experiments, ensuring consistency in architecture and training setup. The key algorithm-specific hyperparameters are listed in Table 3.

| Hyperparameter | —— IQN | —— IQN+MINTO |
|---|---|---|
| Replay Buffer Capacity | 1,000,000 | |
| Batch Size | 32 | |
| Update Horizon ($n$) | 1 | |
| Discount Factor ($\gamma$) | 0.99 | |
| Learning Rate | $5.0 \times 10^{-5}$ | |
| Adam $\epsilon$ | $3.125 \times 10^{-4}$ | |
| Horizon | 27,000 | |
| Architecture Type | CNN | |
| Features | [32, 64, 64, 512] | |
| Epochs | 100 | |
| Training Steps per Epoch | 250,000 | |
| Data to Update | 4 | |
| Initial Samples | 20,000 | |
| Epsilon End | 0.01 | |
| Epsilon Duration | 250,000 | |
| Target Update Frequency ($T$) | 8000 | |

Table 3: Comparison of IQN hyperparameters for $n = 1$ using IQN and IQN+MINTO. Update horizon is set to 1.

## C.3 OFFLINE RL

For the offline reinforcement learning experiments on Atari, we use datasets from RL Unplugged[1] (Gulcehre et al., 2020), which provide standardized and diverse benchmarks. The implementation is built on a stable and well-tested codebase[2] to ensure reproducibility and fair comparison, released by Vincent et al. (2025a). The codebase has been released. All methods are run with their default hyperparameters, and the most important settings are reported in Table 4.

| Hyperparameter | CNN | | IMPALA | |
|---|---|---|---|---|
| | — CQL | — CQL+MINTO | — CQL | — CQL+MINTO |
| Dataset Size | 5,000,000 | | | |
| Batch Size | 32 | | | |
| Update Horizon | 1 | | | |
| Discount Factor ($\gamma$) | 0.99 | | | |
| Epochs | 100 | | | |
| Learning Rate | $5 \times 10^{-5}$ | | | |
| Adam ($\epsilon$) | $5 \times 3.125^{-4}$ | | | |
| Training Steps per Epoch | 62,500 | | | |
| Tradeoff Factor ($\alpha$) | 0.1 | | | |
| Target Update Frequency ($T$) | 2000 | | | |
| Layer Norm | no | | yes | |

Table 4: Comparison of CQL and CQL+MINTO hyperparameters for the two different network architectures CNN and IMPALA.

## C.4 ONLINE RL AND CONTINUOUS CONTROL

For our continuous-control experiments with online reinforcement learning, we adopt SimbaV1 and SimbaV2. The implementation is based on the official SimbaV2 codebase[3] (Lee et al., 2025), ensuring consistency with the original work. All experiments use the default hyperparameters provided by the authors, with the most relevant ones summarized in Tables 5 and 6.

| Hyperparameter | — SimbaV1, — SimbaV1+MINTO | | |
|---|---|---|---|
| | DMC-Hard | HumanoidBench | MuJoCo |
| Discount Factor ($\gamma$) | 0.99 | | 0.995 |
| Learning Rate | $1.0 \times 10^{-4}$ | | |
| Weight Decay | 0.01 | | |
| Target ($\tau$) | 0.005 | | |
| Update Horizon ($n$) | 1 | | |
| Temperature Initial Value | 0.01 | | |
| Temperature Target Entropy | $-0.5 \times |\mathcal{A}|$ | | |
| Batch Size | 256 | | |
| Buffer Max Length | 1,000,000 | | |
| Buffer Min Length | 5,000 | | |
| Num Train Envs | 1 | | |
| Action Repeat | 2 | | 1 |
| Max Episode Steps | 1000 | | |

Table 5: Comparison of SimbaV1 hyperparameters across DMC-Hard, HumanoidBench, and Mujoco locomotion environments. Identical values are merged.

---

[1] https://github.com/huihanl/rl_unplugged
[2] https://github.com/slimRL/slimCQL
[3] https://github.com/dojeon-ai/SimbaV2

| Hyperparameter | SimbaV2, SimbaV2+MINTO | | |
| --- | --- | --- | --- |
| | **DMC-Hard** | **HumanoidBench** | **MuJoCo** |
| Actor Shift | 3 | | |
| Critic Shift | 3 | | |
| Critic $v_{\max}$ | 5.0 | | |
| Critic $v_{\min}$ | $-5.0$ | | |
| Critic Num Bins | 101 | | |
| Discount Factor ($\gamma$) | 0.99 | | 0.995 |
| Learning Rate Init | $1.0 \times 10^{-4}$ | | |
| Learning Rate End | $5.0 \times 10^{-5}$ | | |
| Update Horizon ($n$) | 1 | | |
| Target ($\tau$) | 0.005 | | |
| Temperature Initial Value | 0.01 | | |
| Temperature Target Entropy | $-0.5 \times |\mathcal{A}|$ | | |
| Buffer Max Length | 1,000,000 | | |
| Buffer Min Length | 5,000 | | |
| Num Train Envs | 1 | | |
| Action Repeat | 2 | | 1 |
| Max Episode Steps | 1000 | | |

Table 6: Comparison of SimbaV2 hyperparameters across DMC-Hard, Humanoid Bench, and MuJoCo. Identical values are merged for clarity.

# D  INDIVIDUAL RESULTS

## D.1  ONLINE RL AND DISCRETE CONTROL

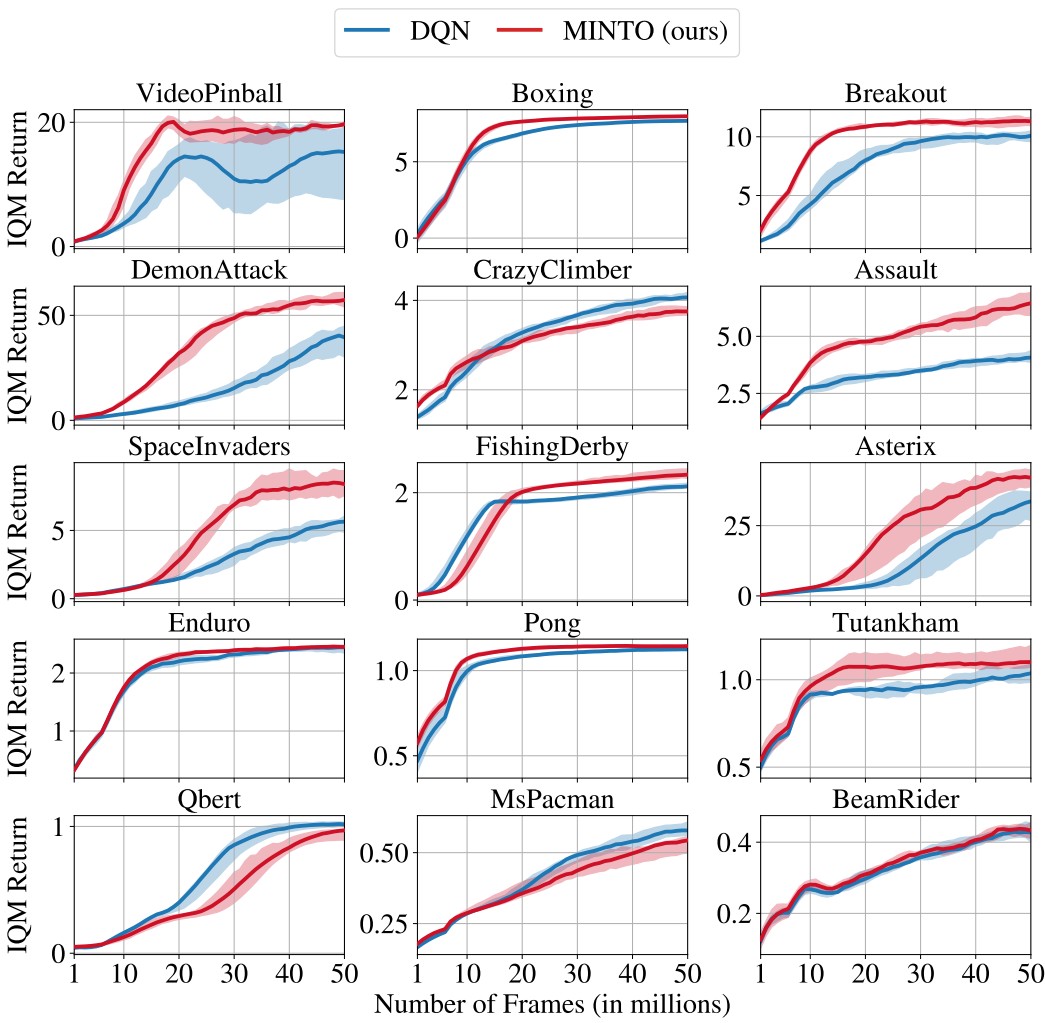

Figure 8: Individual Results of benchmarking MINTO and DQN on 15 Atari games using the IM-PALA architecture with LayerNorm. Reported metrics are interquartile mean (IQM) scores with 95% confidence intervals across 5 seeds per game.

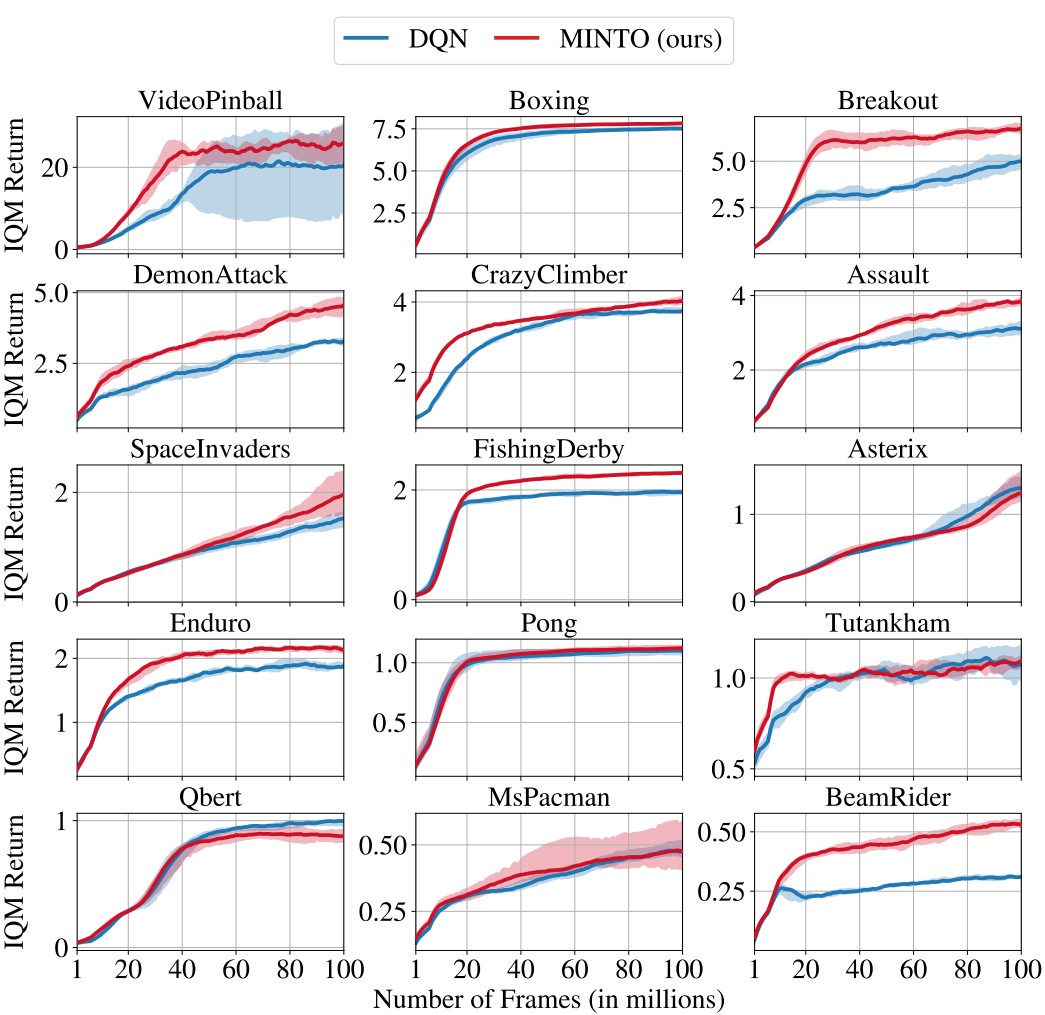

Figure 9: Individual Results of benchmarking MINTO and DQN on 15 Atari games using the CNN network architecture. Reported metrics are interquartile mean (IQM) scores with 95% confidence intervals across 5 seeds per game.

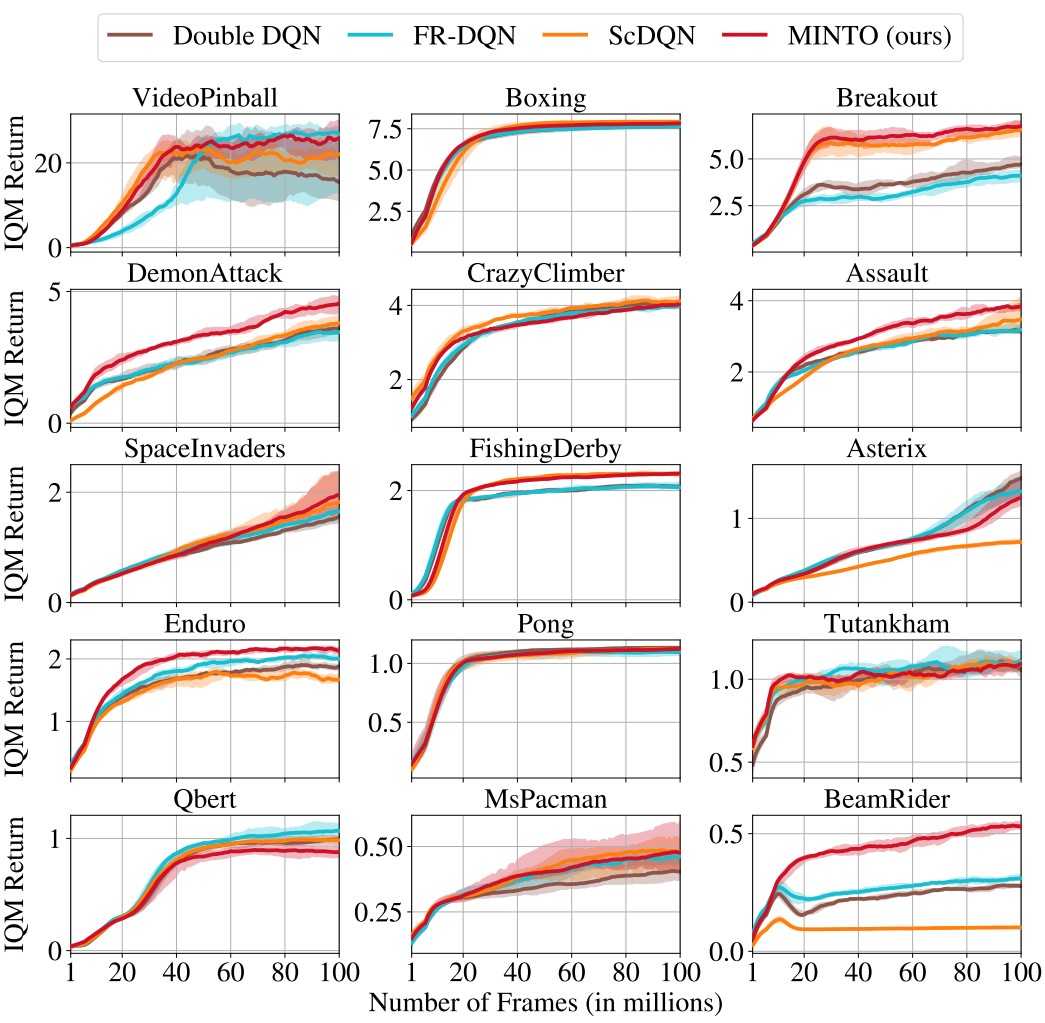

Figure 10: Individual Results of benchmarking Double DQN, FR-DQN, ScDQN, and MINTO on 15 Atari games using the CNN network architecture. Reported metrics are interquartile mean (IQM) scores with 95% confidence intervals across 5 seeds per game.

## D.2 MAXMIN DQN VS. MINTO

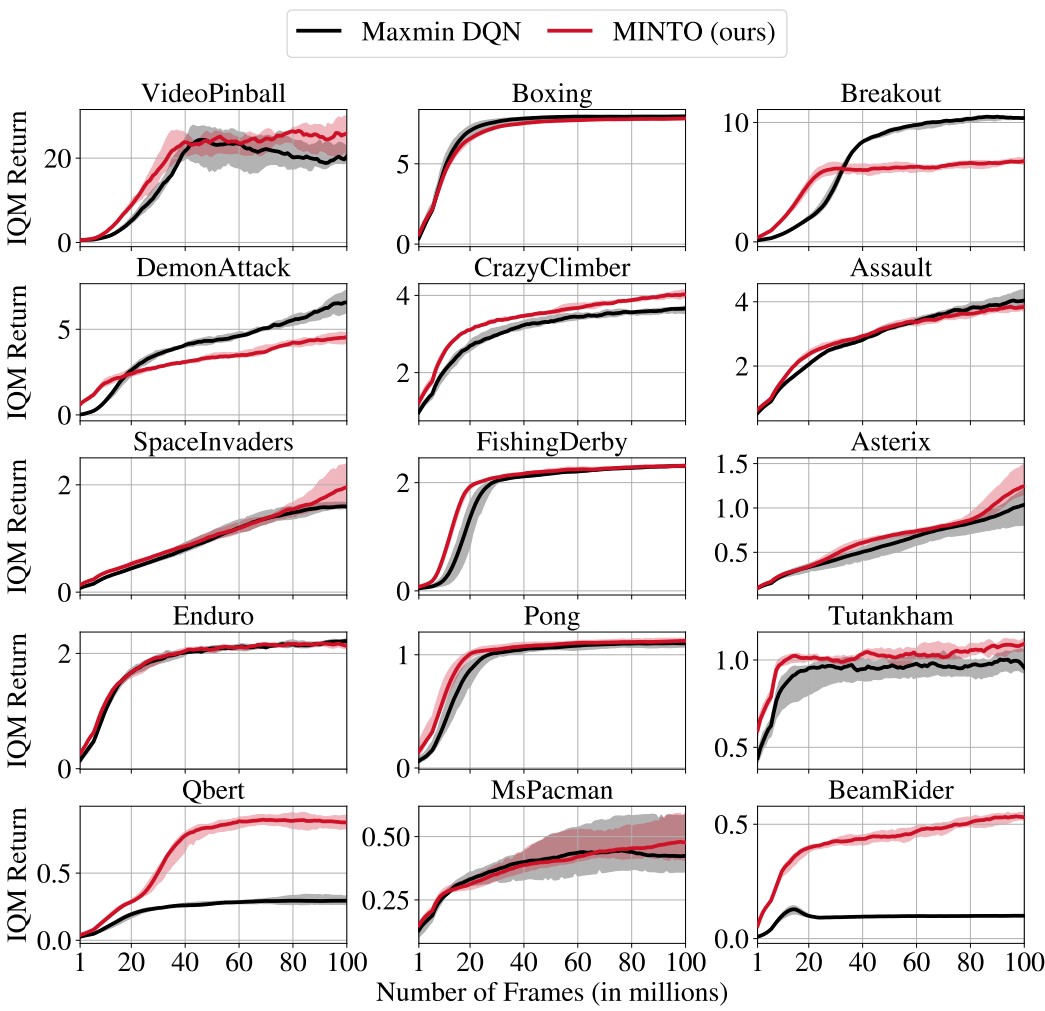

Figure 11: Individual Results of benchmarking Maxmin DQN and MINTO on 15 Atari games using the CNN network architecture. Reported metrics are interquartile mean (IQM) scores with 95% confidence intervals across 5 seeds per game.

## D.3 DISTRIBUTIONAL RL

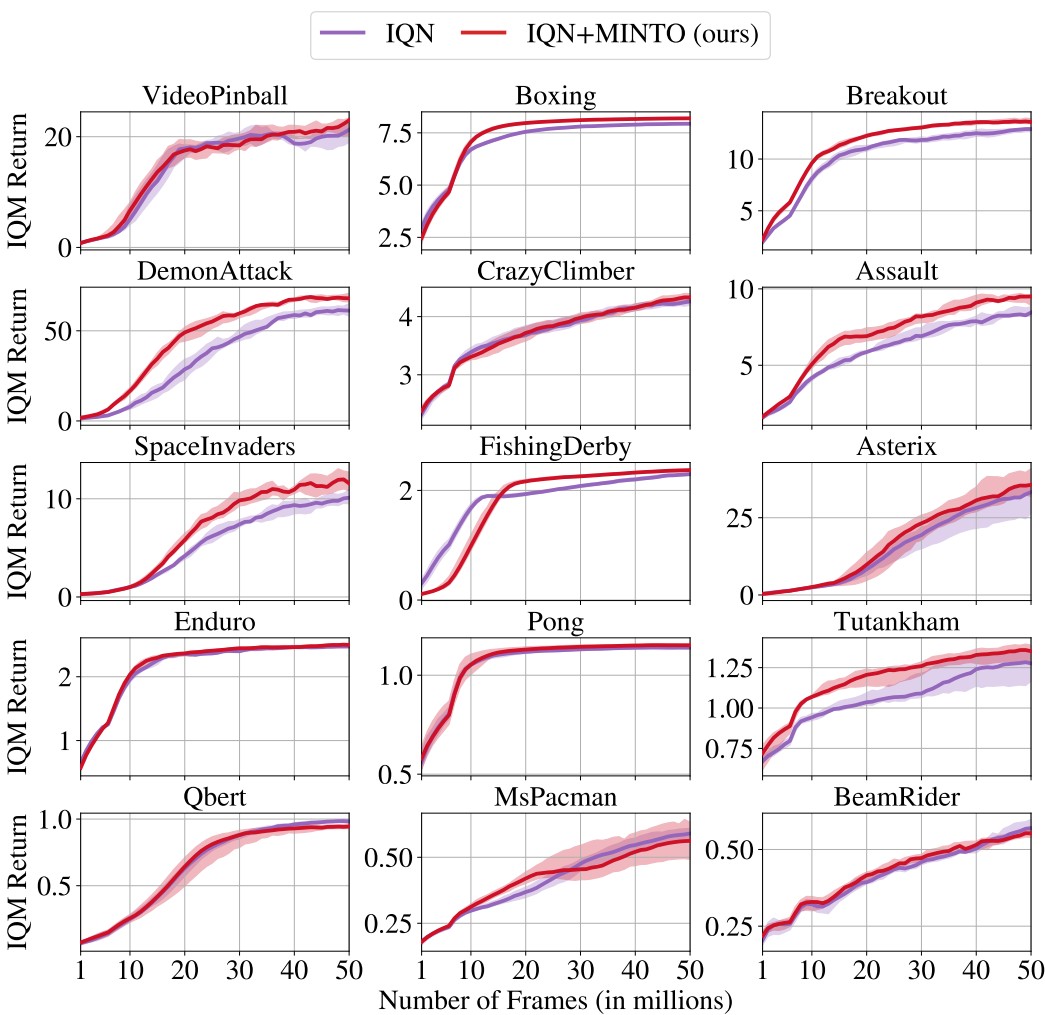

Figure 12: Individual Results of benchmarking IQN and IQN+MINTO on 15 Atari games using the IMPALA architecture with LayerNorm. Reported metrics are interquartile mean (IQM) scores with 95% confidence intervals across 5 seeds per game.

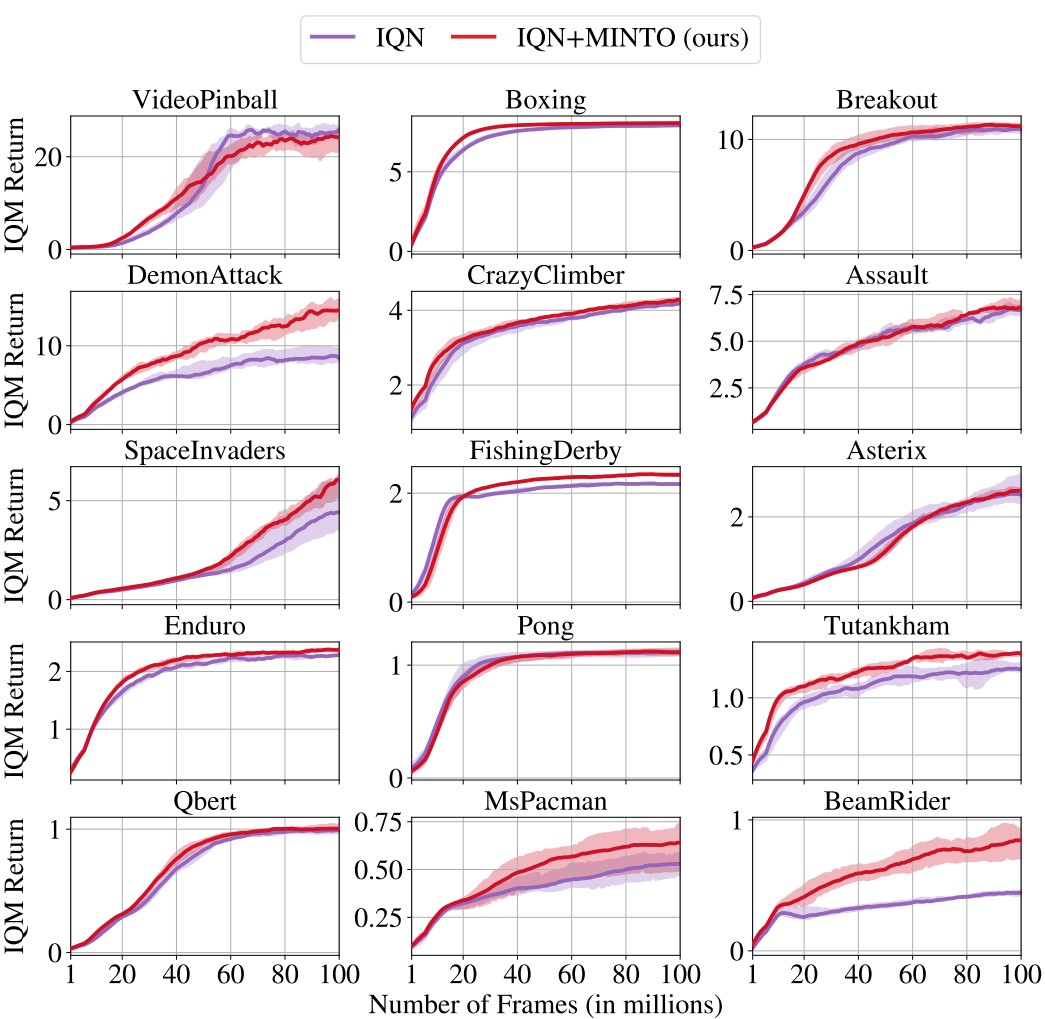

Figure 13: Individual Results of benchmarking IQN and IQN+MINTO on 15 Atari games using the CNN network architecture. Reported metrics are interquartile mean (IQM) scores with 95% confidence intervals across 5 seeds per game.

## D.4 OFFLINE RL

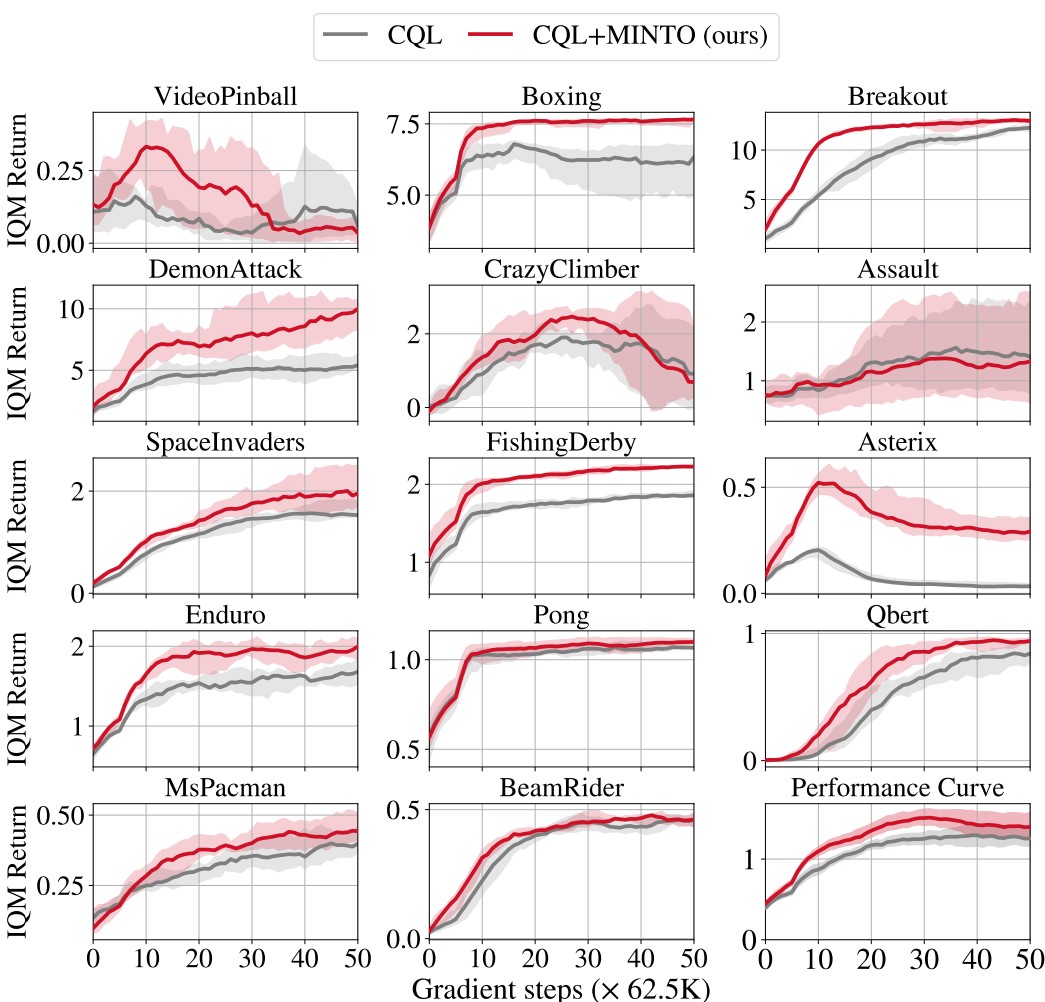

Figure 14: Individual Results of benchmarking CQL and CQL+MINTO on 14 Atari games using the IMPALA architecture with LayerNorm. Reported metrics are interquartile mean (IQM) scores with 95% confidence intervals across 5 seeds per game.

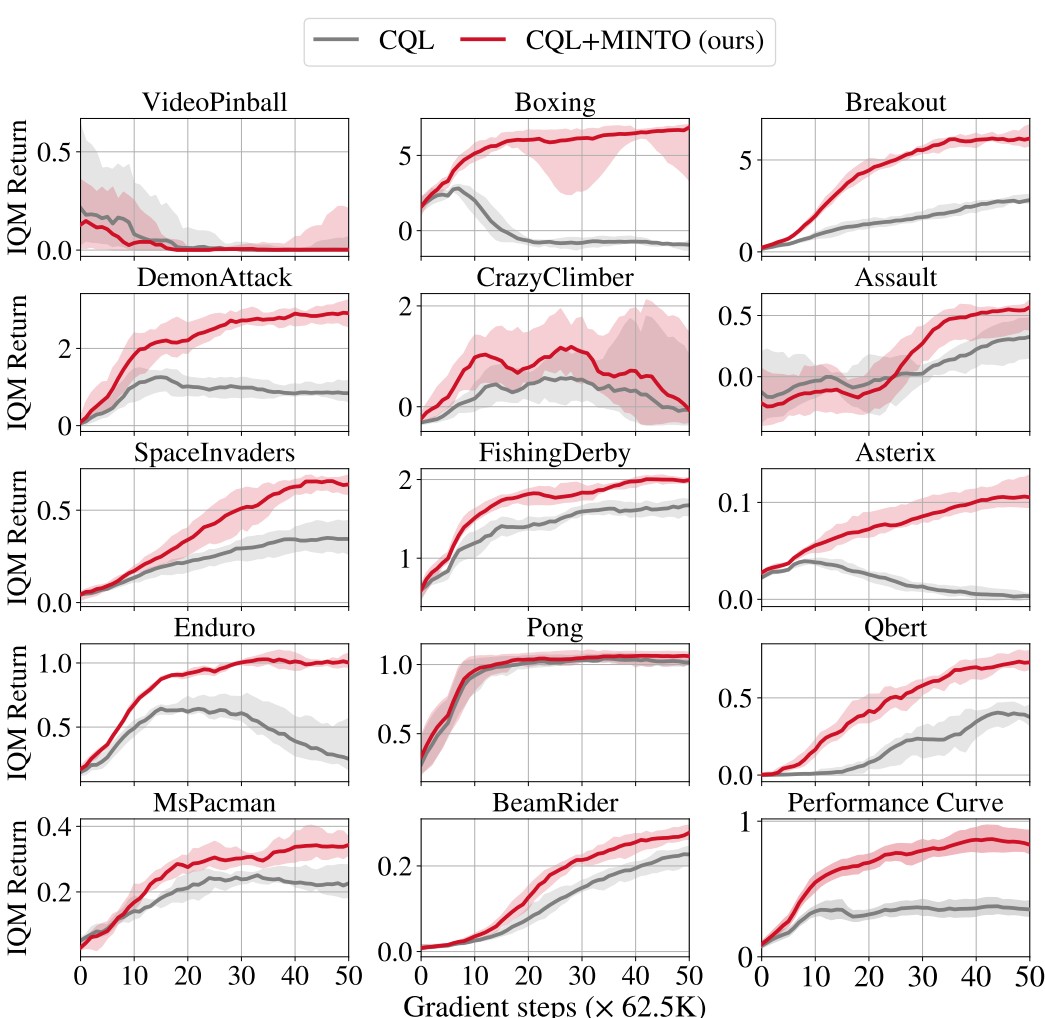

Figure 15: Individual Results of benchmarking CQL and CQL+MINTO on 14 Atari games using the CNN architecture. Reported metrics are interquartile mean (IQM) scores with 95% confidence intervals across 5 seeds per game.

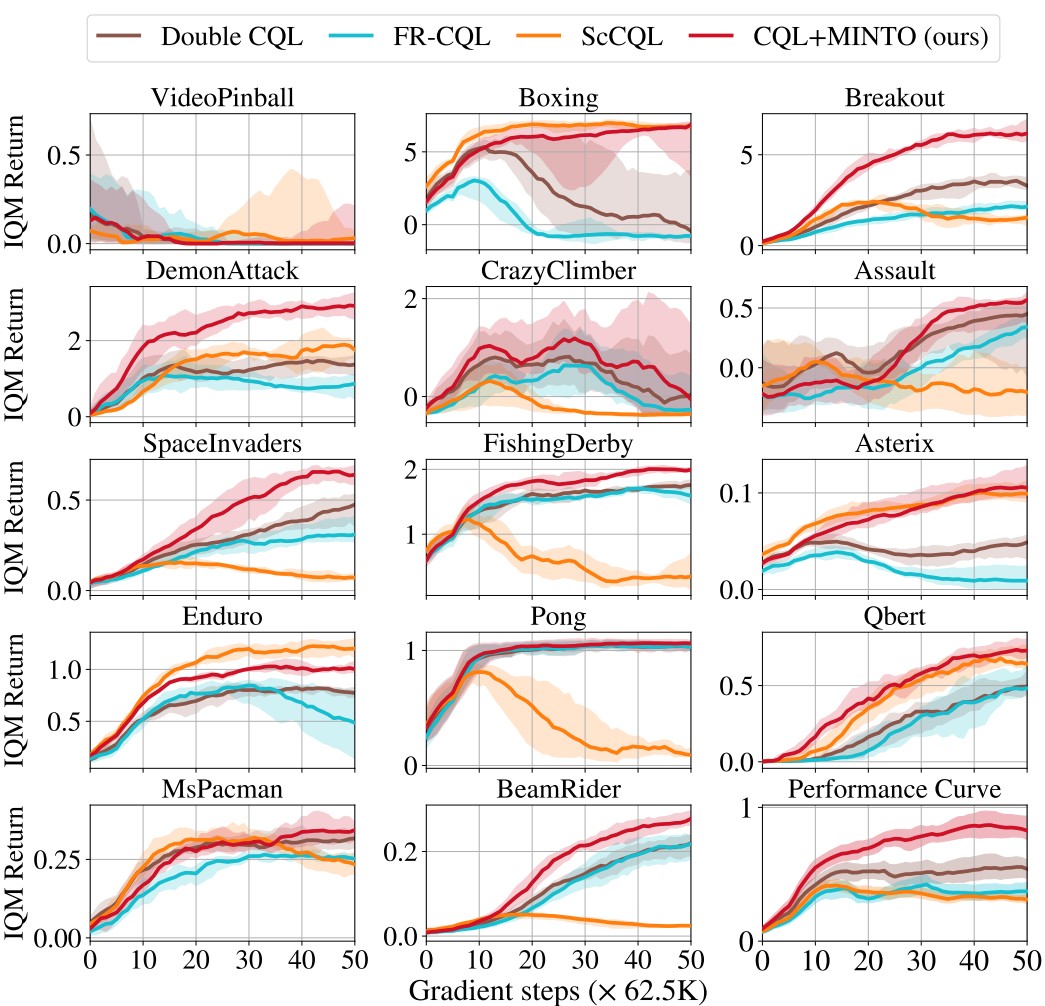

Figure 16: Individual Results of benchmarking Double CQL, FR-CQL, ScCQL, and CQL+MINTO on 14 Atari games using the CNN network architecture. Reported metrics are interquartile mean (IQM) scores with 95% confidence intervals across 5 seeds per game.

## D.5 MAXMIN CQL VS. MINTO

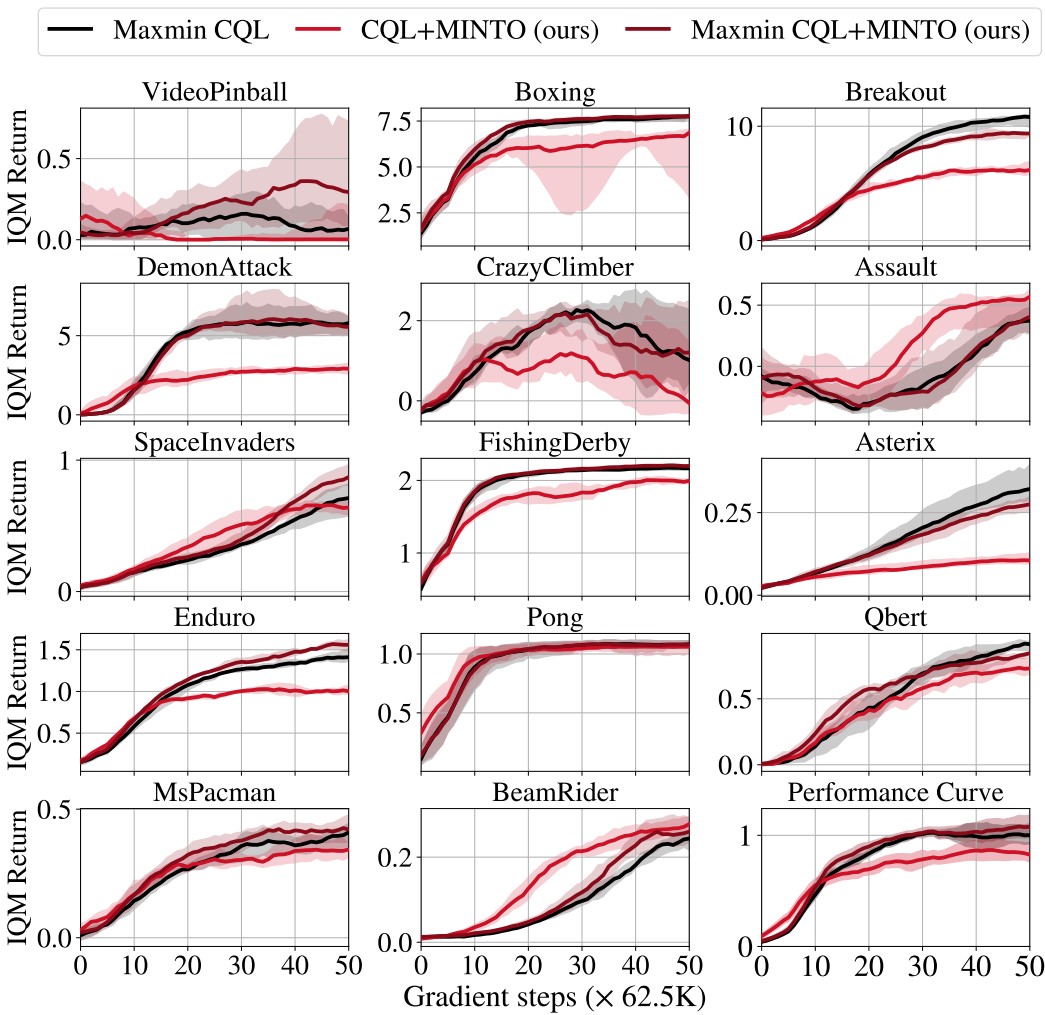

Figure 17: Individual Results of benchmarking Maxmin CQL, CQL+MINTO, and Maxmin CQL+MINTO on 14 Atari games using the CNN network architecture. Reported metrics are interquartile mean (IQM) scores with 95% confidence intervals across 5 seeds per game.

## D.6 ONLINE RL AND CONTINUOUS CONTROL

### D.6.1 SIMBA

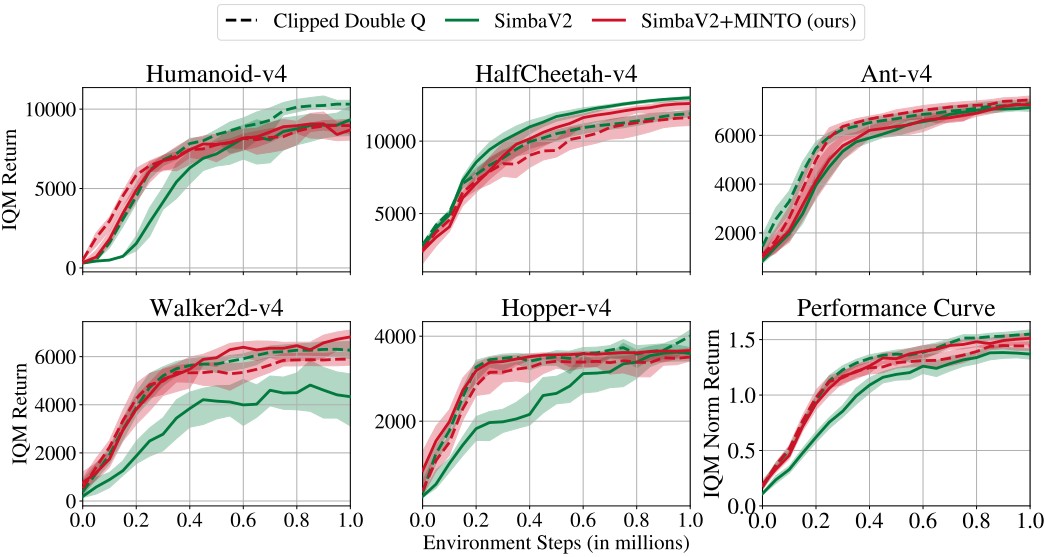

Figure 18: Individual Results of benchmarking SimbaV2 and SimbaV2+MINTO on the 5 MuJoCo environments. Reported metrics are interquartile mean (IQM) scores with 95% confidence intervals across 10 seeds per environment. The last plot (**bottom right**) shows the IQM of the normalized return over all 5 environments.

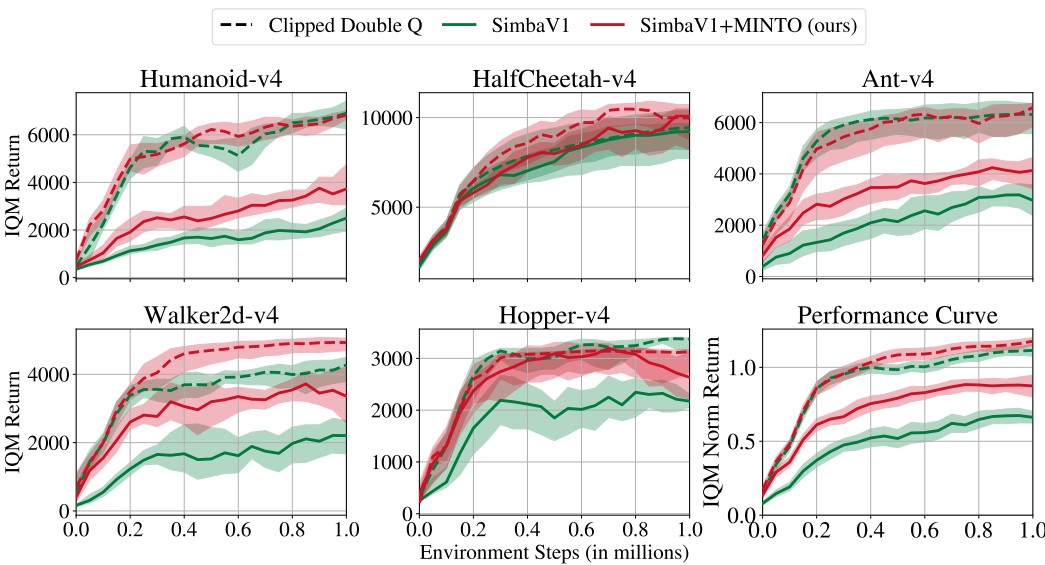

Figure 19: Individual Results of benchmarking SimbaV1 and SimbaV1+MINTO on the 5 MuJoCo environments. Reported metrics are interquartile mean (IQM) scores with 95% confidence intervals across 10 seeds per environment. The last plot (**bottom right**) shows the IQM of the normalized return over all 5 environments.

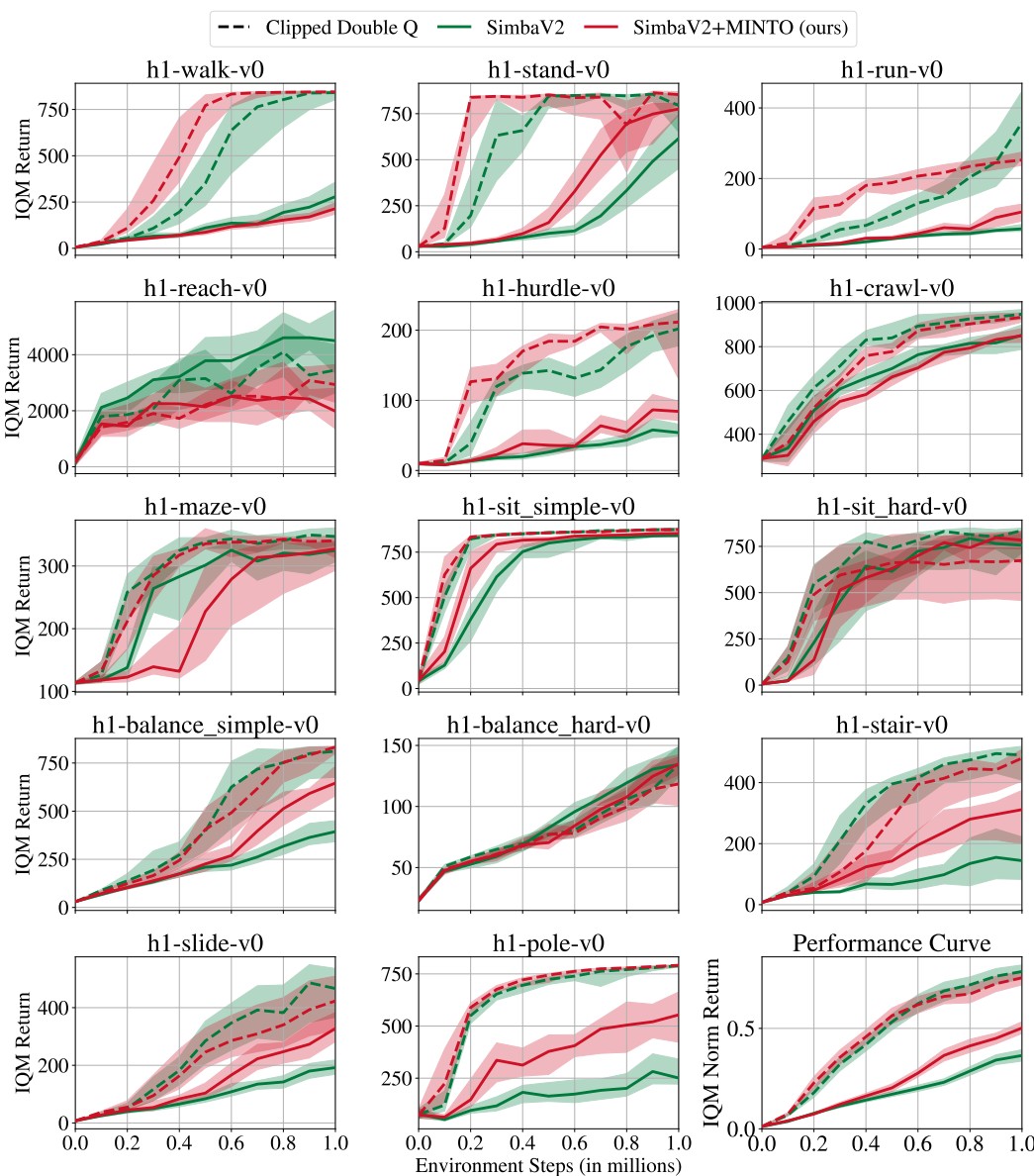

Figure 20: Individual Results of benchmarking SimbaV2 and SimbaV2+MINTO on the 14 Humanoid Bench environments. Reported metrics are interquartile mean (IQM) scores with 95% confidence intervals across 10 seeds per environment. The last plot (**bottom right**) shows the IQM of the normalized return over all 14 environments.

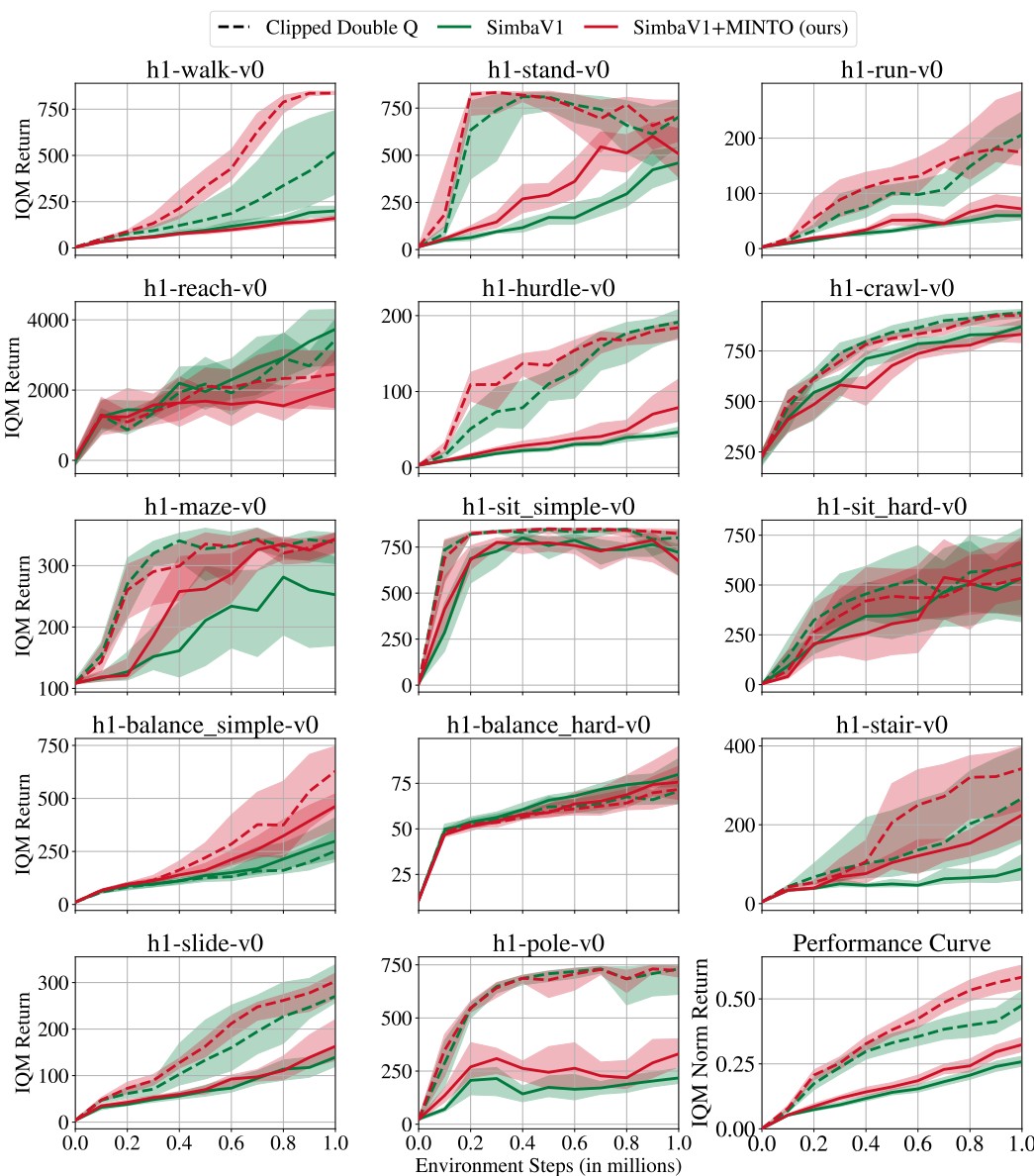

Figure 21: Individual Results of benchmarking SimbaV1 and SimbaV1+MINTO on the 14 Humanoid Bench environments. Reported metrics are interquartile mean (IQM) scores with 95% confidence intervals across 10 seeds per environment. The last plot (**bottom right**) shows the IQM of the normalized return over all 14 environments.

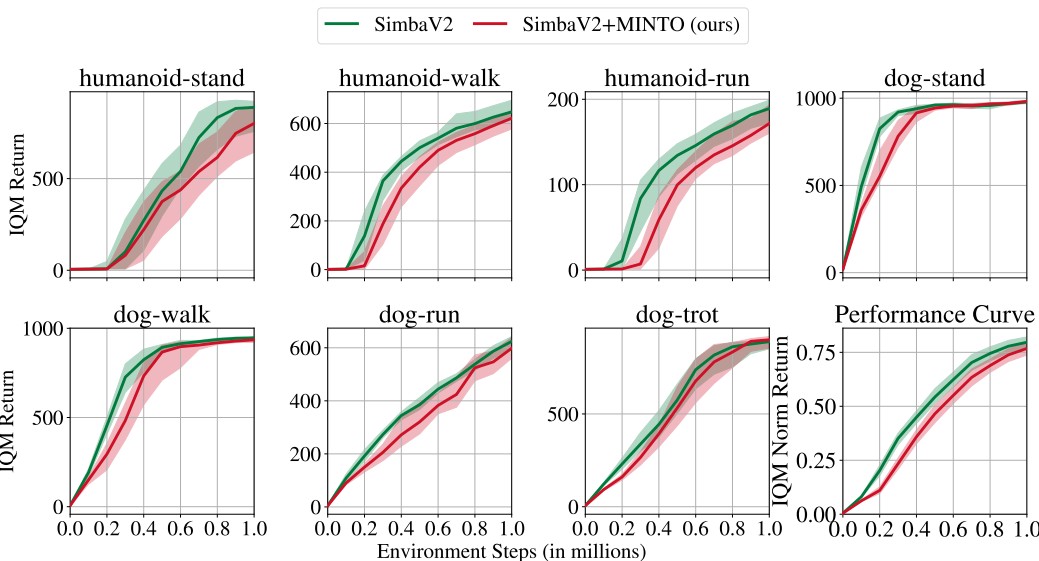

Figure 22: Individual Results of benchmarking SimbaV2 and SimbaV2+MINTO on the 7 DMC-Hard environments. Reported metrics are interquartile mean (IQM) scores with 95% confidence intervals across 10 seeds per environment. The last plot (**bottom right**) shows the IQM of the normalized return over all 7 environments.

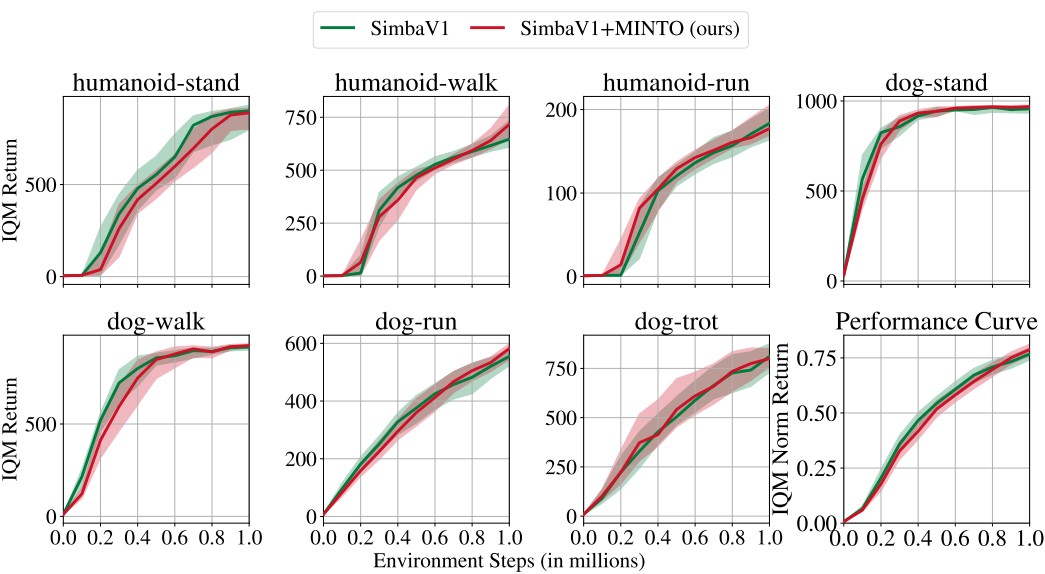

Figure 23: Individual Results of benchmarking SimbaV1 and SimbaV1+MINTO on the 7 DMC-Hard environments. Reported metrics are interquartile mean (IQM) scores with 95% confidence intervals across 10 seeds per environment. The last plot (**bottom right**) shows the IQM of the normalized return over all 7 environments.

### D.6.2  CROSSQ+WN

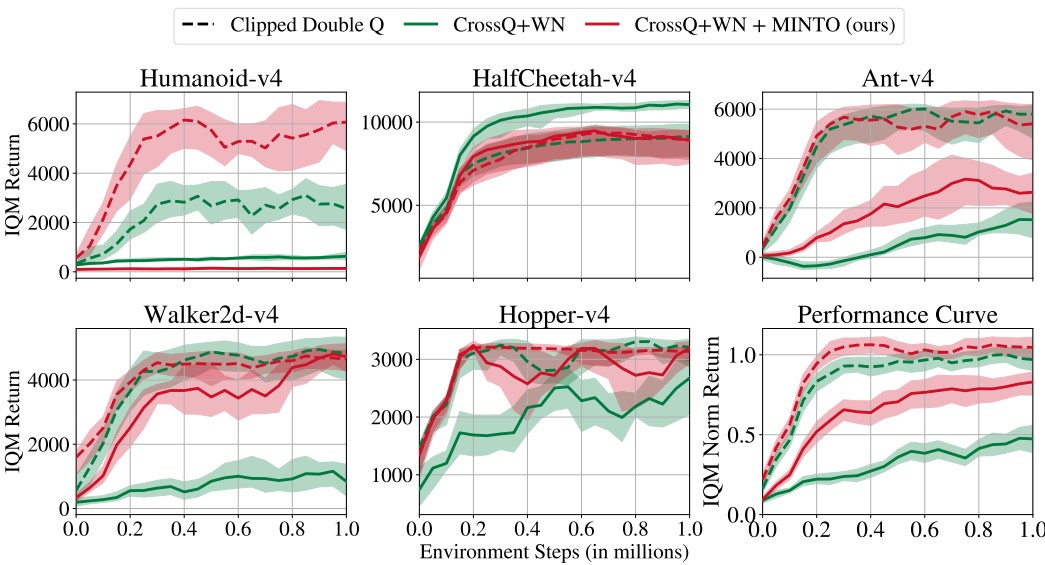

Figure 24: Individual Results of benchmarking CrossQ+WN and CrossQ+WN + MINTO on the 5 MuJoCo environments. Reported metrics are interquartile mean (IQM) scores with 95% confidence intervals across 10 seeds per environment. The last plot (**bottom right**) shows the IQM of the normalized return over all 5 environments.

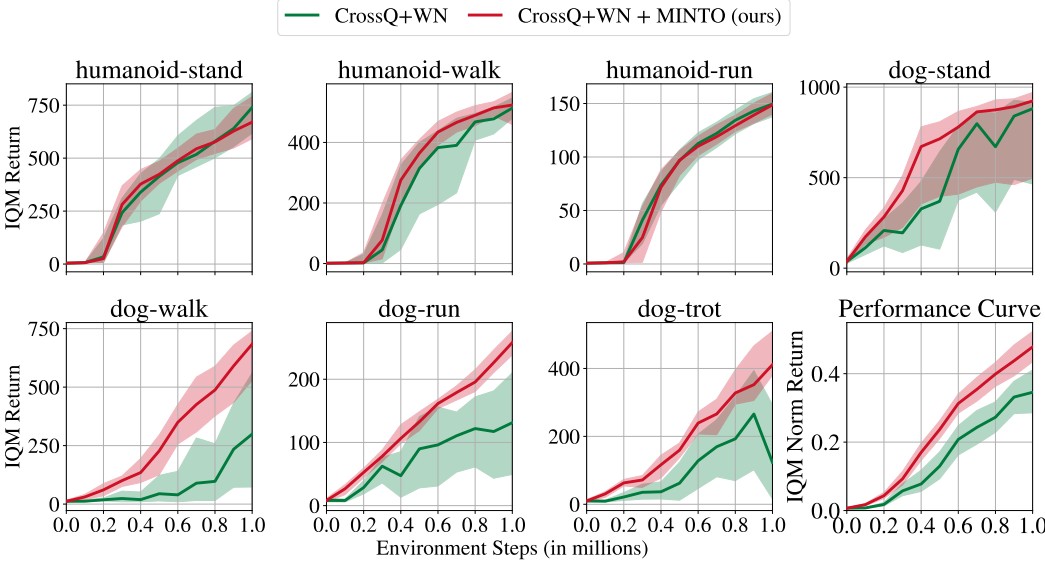

Figure 25: Individual Results of benchmarking CrossQ+WN and CrossQ+WN + MINTO on the 7 DMC-Hard environments. Reported metrics are interquartile mean (IQM) scores with 95% confidence intervals across 10 seeds per environment. The last plot (**bottom right**) shows the IQM of the normalized return over all 7 environments.

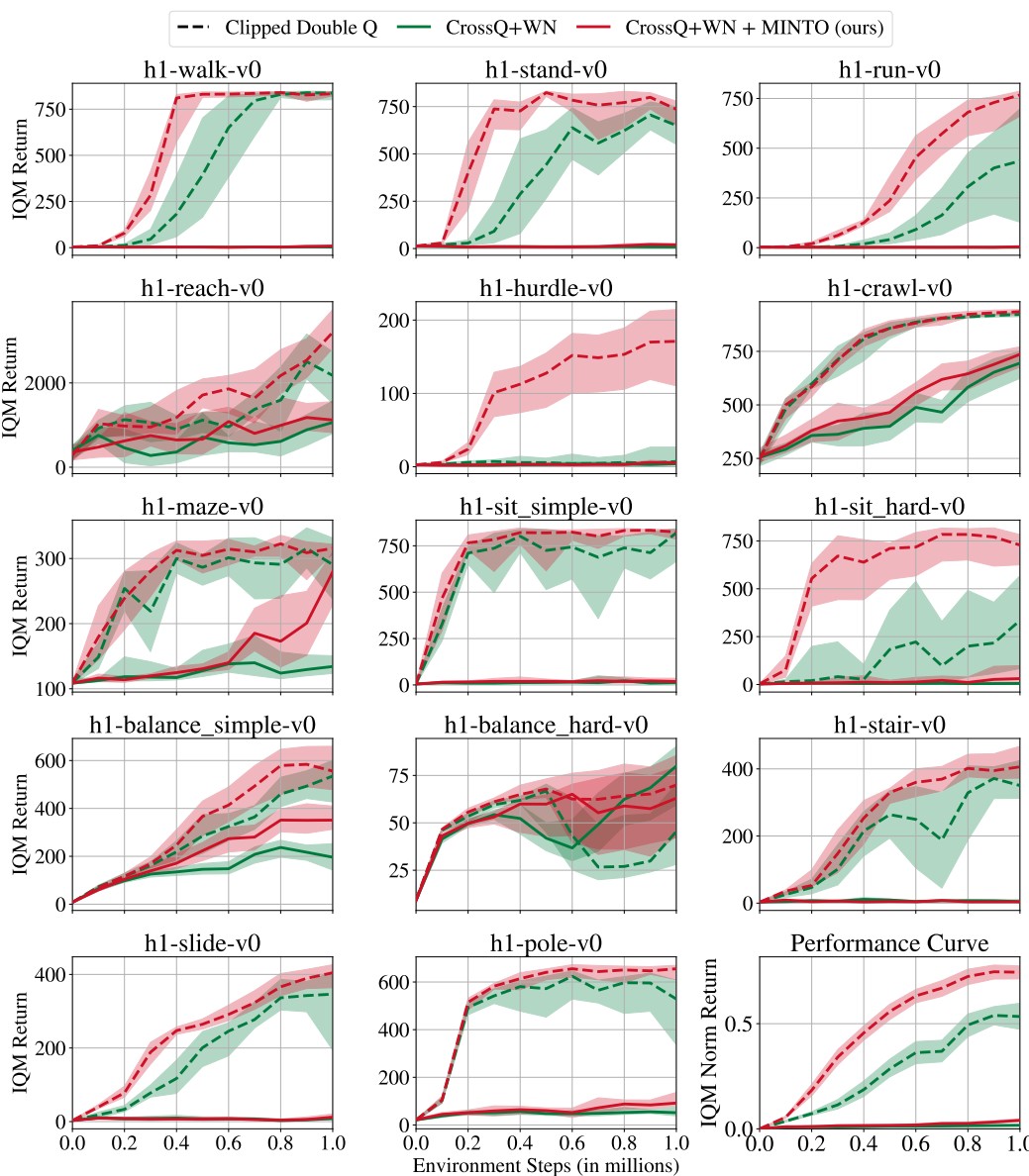

Figure 26: Individual Results of benchmarking CrossQ+WN and CrossQ+WN + MINTO on the 14 Humanoid Bench environments. Reported metrics are interquartile mean (IQM) scores with 95% confidence intervals across 10 seeds per environment. The last plot (**bottom right**) shows the IQM of the normalized return over all 14 environments.

### D.6.3  FUNCTIONAL REGULARIZATION VS. MINTO

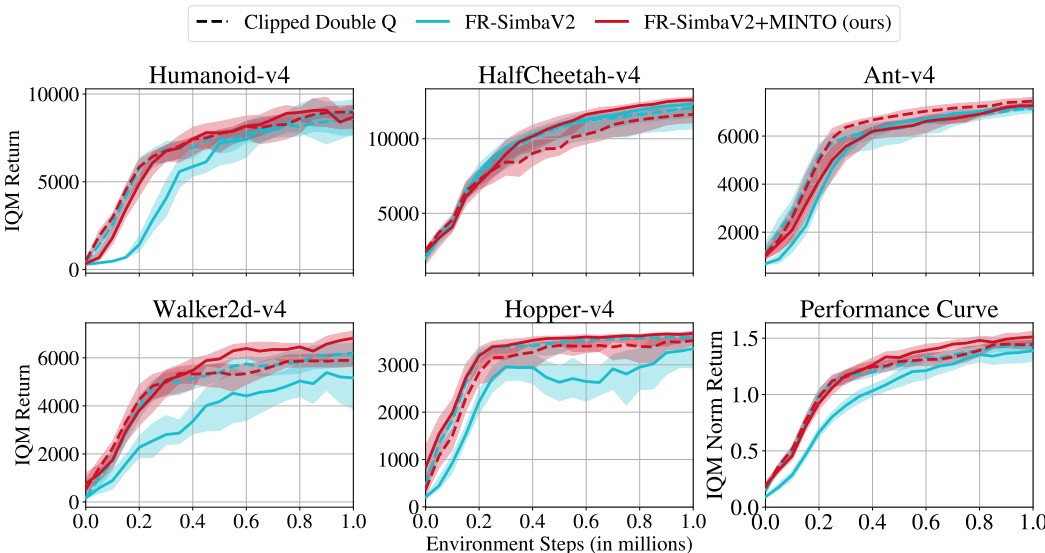

Figure 27: Individual Results of benchmarking FR-SimbaV2 and SimbaV2+MINTO on the 5 Mu-JoCo environments. Reported metrics are interquartile mean (IQM) scores with 95% confidence intervals across 10 seeds per environment. The last plot (**bottom right**) shows the IQM of the normalized return over all 5 environments.

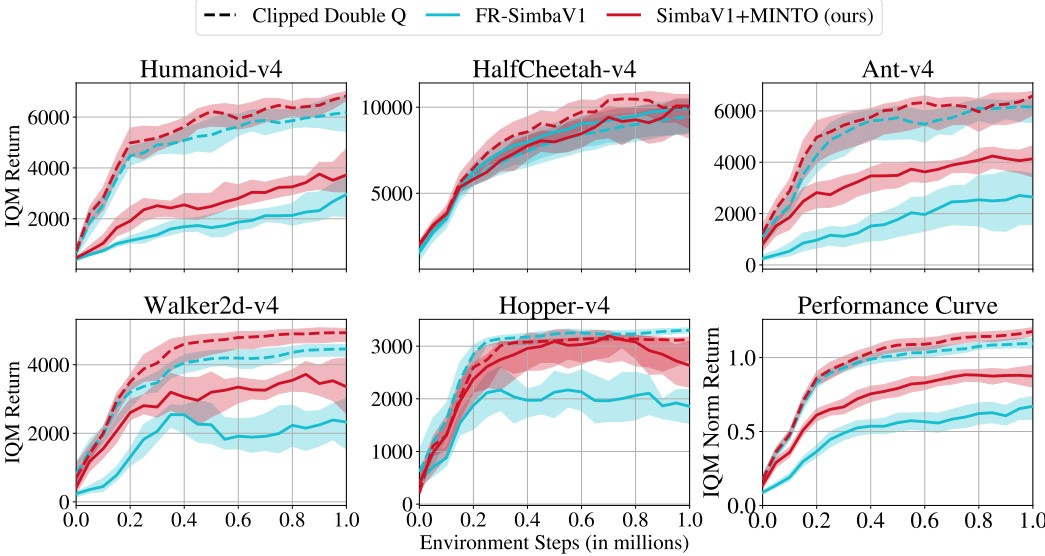

Figure 28: Individual Results of benchmarking FR-SimbaV1 and SimbaV1+MINTO on the 5 Mu-JoCo environments. Reported metrics are interquartile mean (IQM) scores with 95% confidence intervals across 10 seeds per environment. The last plot (**bottom right**) shows the IQM of the normalized return over all 5 environments.

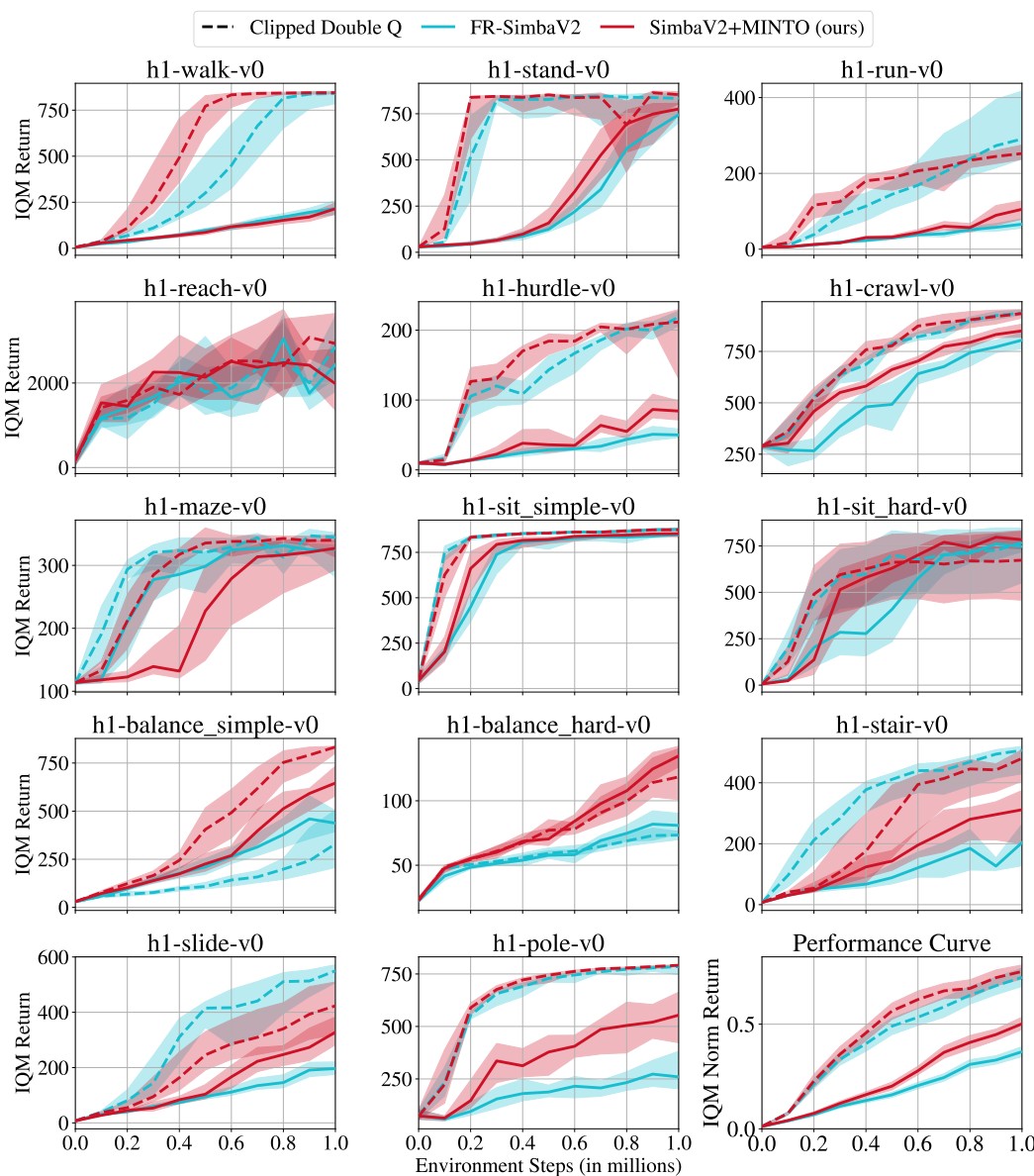

Figure 29: Individual Results of benchmarking FR-SimbaV2 and SimbaV2+MINTO on the 14 Humanoid Bench environments. Reported metrics are interquartile mean (IQM) scores with 95% confidence intervals across 10 seeds per environment. The last plot (**bottom right**) shows the IQM of the normalized return over all 14 environments.

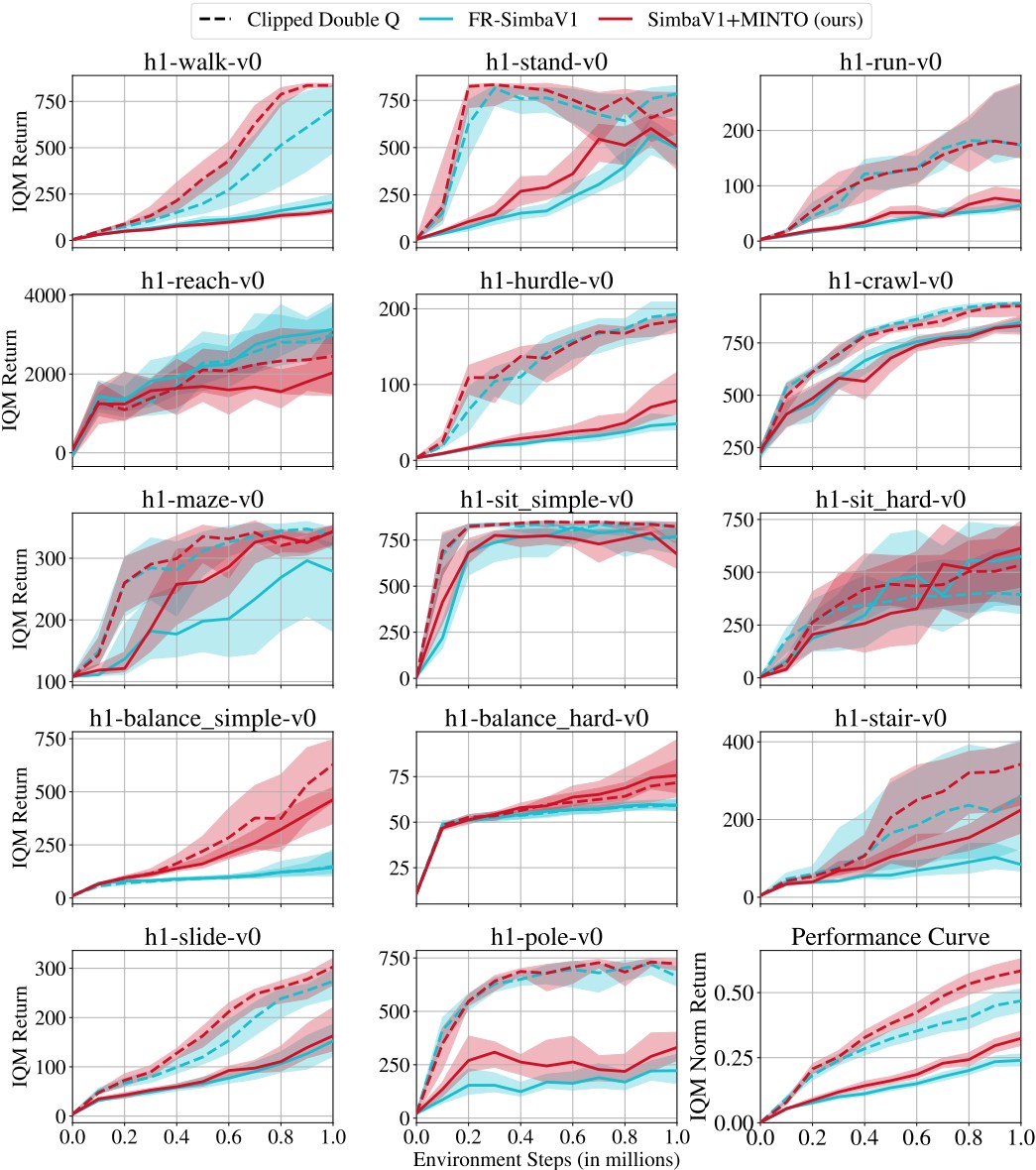

Figure 30: Individual Results of benchmarking FR-SimbaV1 and SimbaV1+MINTO on the 14 Humanoid Bench environments. Reported metrics are interquartile mean (IQM) scores with 95% confidence intervals across 10 seeds per environment. The last plot (**bottom right**) shows the IQM of the normalized return over all 14 environments.

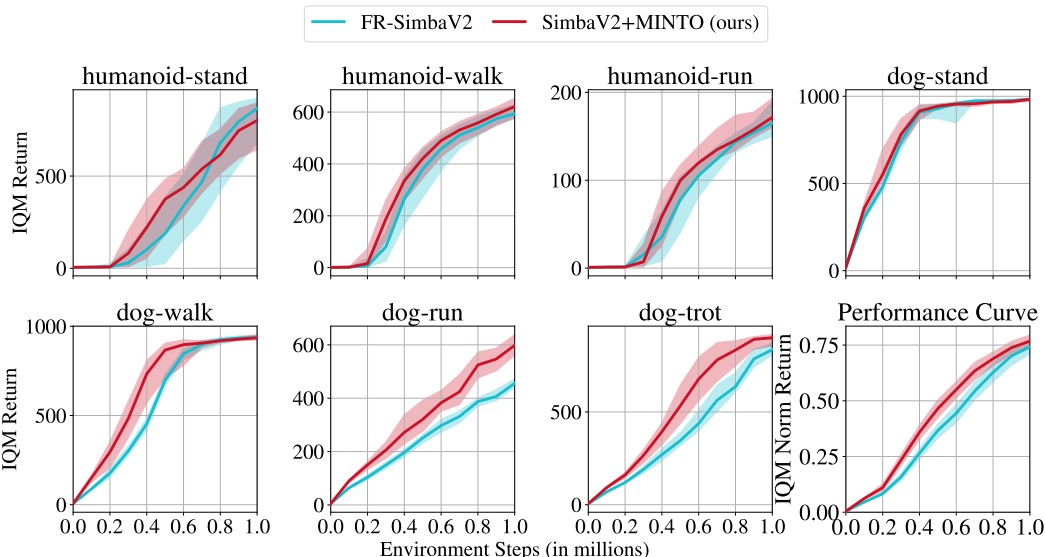

Figure 31: Individual Results of benchmarking FR-SimbaV2 and SimbaV2+MINTO on the 7 DMC-Hard environments. Reported metrics are interquartile mean (IQM) scores with 95% confidence intervals across 10 seeds per environment. The last plot (**bottom right**) shows the IQM of the normalized return over all 7 environments.

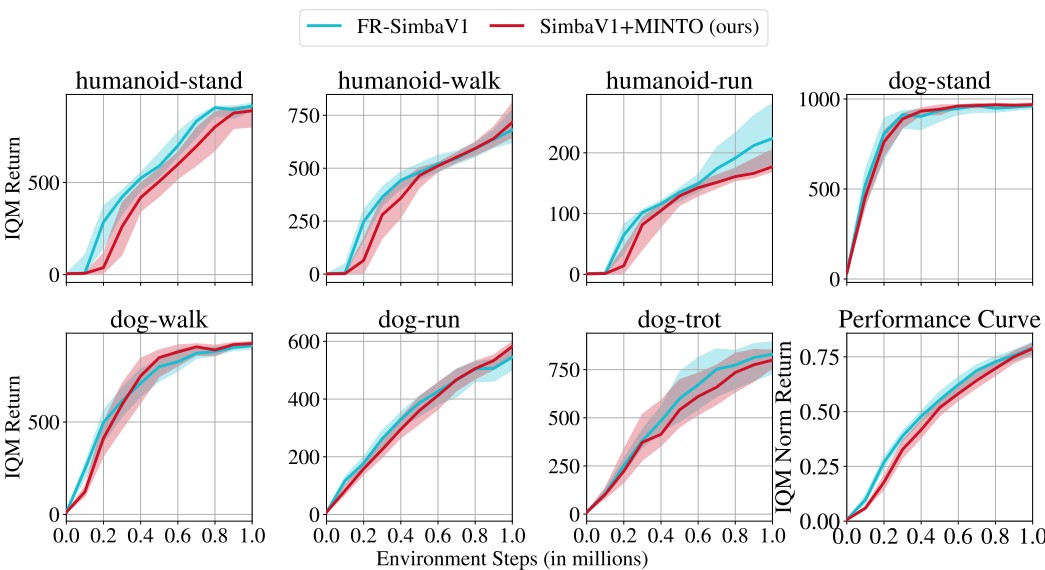

Figure 32: Individual Results of benchmarking FR-SimbaV1 and SimbaV1+MINTO on the 7 DMC-Hard environments. Reported metrics are interquartile mean (IQM) scores with 95% confidence intervals across 10 seeds per environment. The last plot (**bottom right**) shows the IQM of the normalized return over all 7 environments.

### D.7    ABLATION ON TARGET OPERATORS

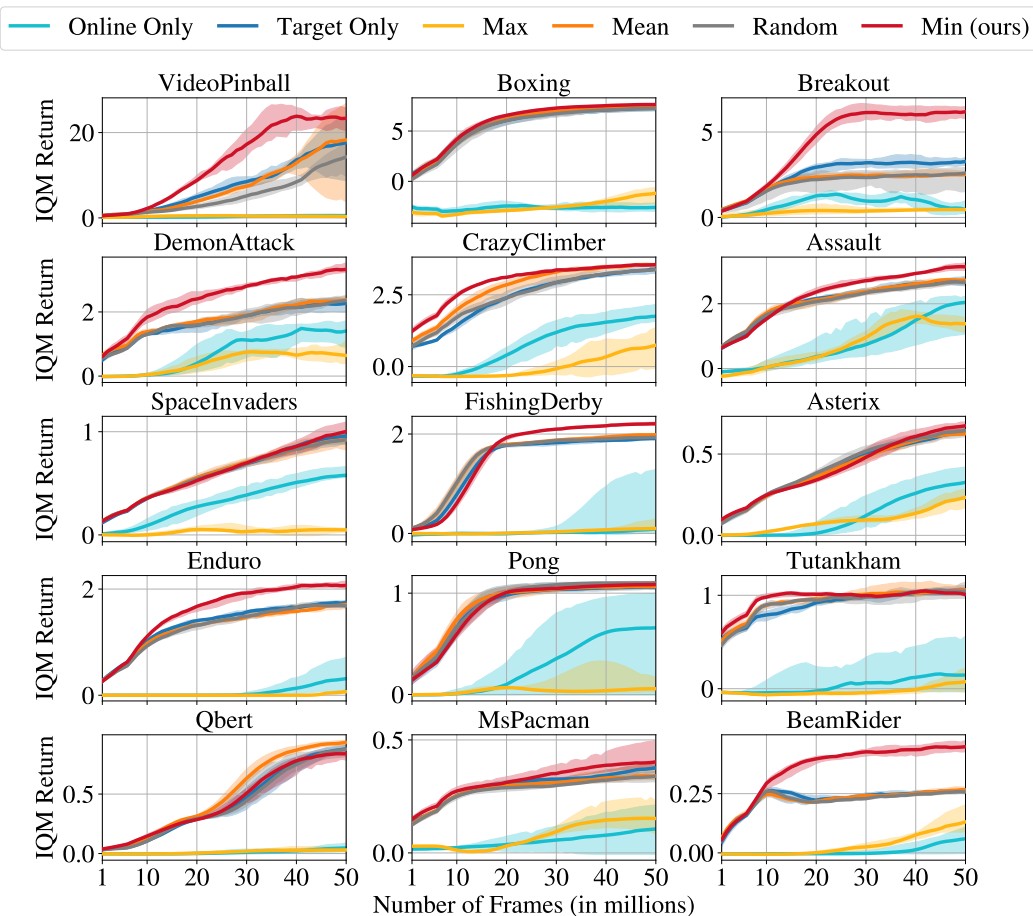

Figure 33: Individual Results of benchmarking the Minimum operator of MINTO against other potential operators on 15 Atari games using the CNN architecture. Reported metrics are interquartile mean (IQM) scores with 95% confidence intervals across 5 seeds per game.

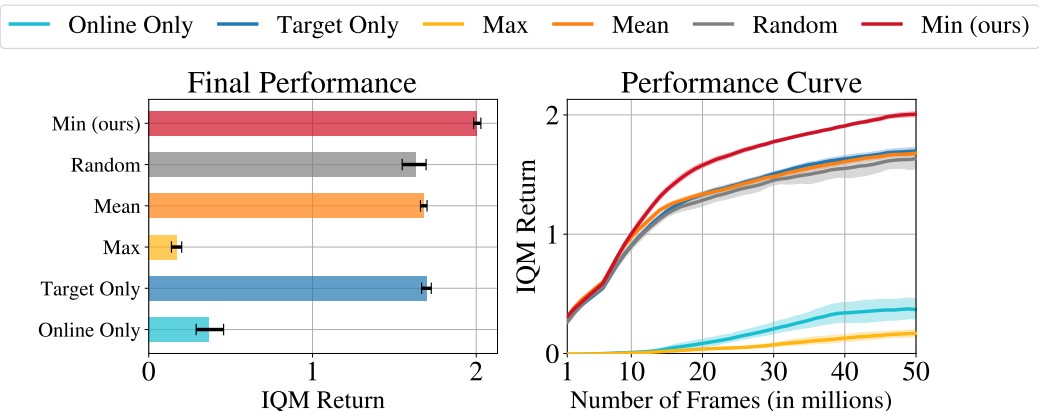

Figure 34: Cumulative results of benchmarking the Minimum operator of MINTO against other potential operators on 15 Atari games using the CNN architecture. Both figures show the interquartile mean (IQM) scores with 95% confidence intervals of the final performance of each operator. **Left:** The final performance of each operator in a bar chart. **Right:** The performance curve for each operator over 50 million frames.

# E    ANALYSIS OF VALUE ESTIMATION ERROR AND BIAS

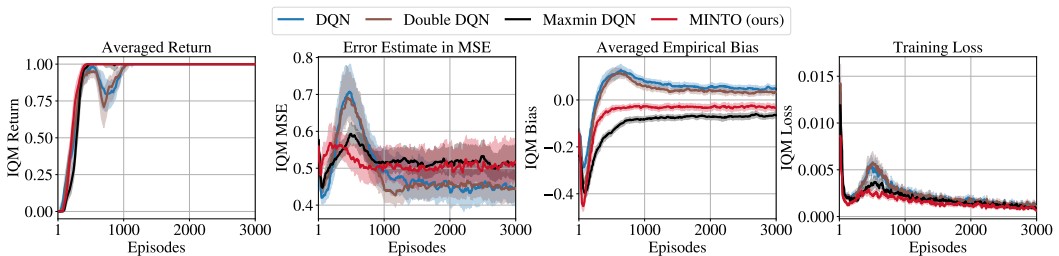

Figure 35: Results of benchmarking DQN, Double DQN, Maxmin DQN, and MINTO on the Car on Hill environment, showing the Average Return, Value Estimation Error in Mean Squared Error (MSE), Averaged Empirical Bias, and Training Loss. Reported metrics are interquartile mean (IQM) scores with 95% confidence intervals across 100 seeds.

In this section, we examine how several value-based RL algorithms perform on the Car-on-the-Hill control task. Our analysis centers on their estimation biases, learning stability, and overall performance, as shown in Fig. 35. We further aim to evaluate whether MINTO's use of recent online estimates enables faster and more stable learning. We also examine the degree to which MINTO exhibits estimation bias when compared with the corresponding baseline algorithms. In particular, Fig. 36 illustrates the instability and severe overestimation bias that arise when relying solely on online estimates for computing the regression target.

## E.1    EXPERIMENTAL SETUP

We consider the *Car-on-the-Hill* environment from Ernst et al. (2005), modeled as a Markov decision process with continuous state space $s = (x, v)$ (position and velocity) and discrete action space $\mathcal{A} = \{0, 1\}$. Episodes terminate upon either reaching the goal region (reward $+1$) or entering a failure region (reward $-1$), with intermediate rewards equal to zero. The episode is limited to 100 steps with a discount factor $\gamma = 0.95$.

To obtain the true value function $Q^\star$, we follow the procedure in Ernst et al. (2005) by computing an exact-lookup approximation on a fixed evaluation grid. For each state–action pair $(s, a)$, we simulate the deterministic transition and, if the successor state is non-terminal, we perform a breadth-first search (BFS) over future trajectories up to a maximum depth $K = 50$. If a success terminal state is discovered at depth $k$, we assign the discounted return $\gamma^{k-1}$; if only failure is reachable within depth $K$, we assign $-\gamma^{k-1}$. This procedure yields a signed, discounted value that we treat as the ground-truth target $Q^\star(s, a)$.

For evaluation, we construct a $10 \times 10$ grid over the state bounds provided by the environment. For each grid point, both actions are considered, yielding 200 fixed test states. The corresponding ground-truth values $Q^\star(s, a)$ are precomputed once and loaded during training.

## E.2    ALGORITHMS AND HYPERPARAMETERS

We compare four value-based deep RL methods: DQN (Mnih et al., 2013), Double DQN (Van Hasselt et al., 2016), Maxmin DQN (Lan et al., 2020), and our approach, MINTO. All agents use a fully connected neural network with two hidden layers of 64 units each and ReLU activations. We train the networks using the Adam optimizer with a learning rate of $10^{-3}$ and mini-batches of size 64 sampled from a replay buffer of capacity 100 000 transitions. A warm-up of 1 000 environment steps is used before performing gradient updates. Target networks are updated every 1 000 environment steps using a hard update.

Exploration follows an $\varepsilon$-greedy strategy, where $\varepsilon$ is annealed linearly from 1.0 to 0.05 over the first 50 000 steps, and kept fixed afterwards. All agents are trained for 3 000 episodes. After each episode, we evaluate the current Q-network on the fixed test grid and compute: (i) Value Estimation Error in Mean Squared Error (MSE) $\mathbb{E}[(\hat{Q}(s, a) - Q^\star(s, a))^2]$, (ii) Averaged Empirical Bias $\mathbb{E}[\hat{Q}(s, a) -$

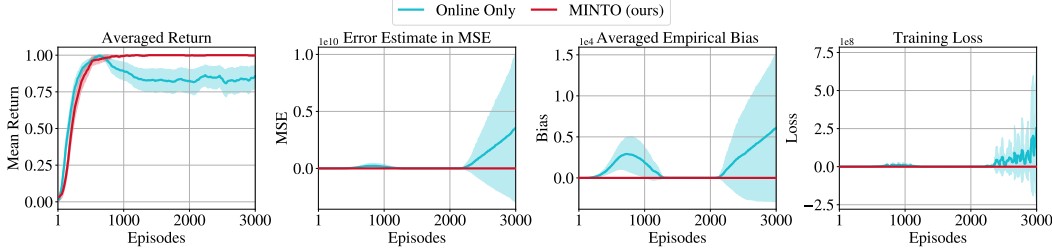

Figure 36: Results of benchmarking Online Only baseline, and MINTO on the Car on Hill environment, showing the Average Return, Value Estimation Error in Mean Squared Error (MSE), Averaged Empirical Bias, and Training Loss. Reported metrics are mean scores with 95% confidence intervals across 100 seeds.

$Q^\star(s, a)$]. Policy performance is monitored by logging episodic Average Return during training and evaluating the greedy policy (with $\varepsilon = 0$) every 20 episodes.

### E.3    DISCUSSION

As anticipated, DQN exhibits a strong overestimation tendency, while Double DQN mitigates this effect only moderately (see Fig. 35). Maxmin DQN, which relies on the minimum over two largely independent value estimates, shows marked underestimation. In contrast, MINTO produces substantially lower overall bias without the severe downward bias characteristic of Maxmin DQN. Importantly, the bias overshoot commonly observed in DQN and Double DQN is absent in MINTO.

While MINTO's estimation error is broadly comparable to that of the baseline methods, it displays a slight increase in variance, an expected outcome of incorporating an online component into the target calculation. Nevertheless, MINTO solves the task faster than all other methods and does so without exhibiting learning instability, despite incorporating online estimates into the target (see Fig. 35). Although Car-on-the-Hill is a deterministic environment with minimal stochasticity, MINTO still performs robustly and avoids the excessive conservatism normally associated with strong underestimation. Finally, in line with Schnell et al. (2025), we observe a similar pattern of training loss increase followed by decrease for the discussed methods, and notably the pattern persists for MINTO but at a lower amplitude.

On the other hand, we demonstrate the effect of relying solely on online estimates when computing the regression target. Fig. 36 highlights the severe overestimation bias that arises from this approach. As a consequence, the value estimation error increases significantly, resulting in degraded performance. The training dynamics, as illustrated by the loss curve, further reveal pronounced instability during learning. Overall, these results validate the design of MINTO, which mitigates such overestimation bias and stabilizes learning by selecting the minimum estimate between the online and target networks when computing the regression target.

