# OpenReview forum: "Use the Online Network If You Can: Towards Fast and Stable Reinforcement Learning"
_ICLR.cc/2026/Conference — ICLR 2026 Poster_

### Official Review · Reviewer_t6g5 · 2025-10-15

**Soundness:** 3
**Presentation:** 3
**Contribution:** 2
**Rating:** 4
**Confidence:** 3

**Summary:**

This paper proposes MINTO, a novel update rule for deep reinforcement learning that computes regression targets as the minimum estimate between the online and target networks. MINTO is designed to address the stability-efficiency trade-off inherent in standard RL, where target networks help stabilize learning but slow down convergence, while online networks foster efficiency but risk instability. Experiments on online, offline, discrete, and continuous RL benchmarks such as Atari games and robotic control tasks demonstrate consistent and significant improvements in sample efficiency and final performance over various baselines.

**Strengths:**

1. The paper is well-motivated and the minimum-operator is easy to integrate into existing RL algorithms.

2. The framework is validated on a wide set of benchmarks (Atari, MuJoCo, offline and online RL) and compared to several baselines, demonstrating consistent gains in both stability and sample efficiency.

3. The modifications require minimal changes to underlying algorithms and introduce only a tiny computational cost, making practical adoption straightforward.

**Weaknesses:**

1. The main evaluation is conducted against SimbaV1 and SimbaV2, which appear to be less strong baselines. This certainly raises the question whether this method can generalize to strong baselines such as Rainbow.

2. The exclusive reliance on the minimum operator can be too conservative, possibly causing underestimation and inhibiting exploration in low-noise or optimistic environments.

3. The code is not attached and will be released upon acceptance. This raises certain concerns.

**Questions:**

1. Have you studied the impact of potential underestimation? While you claim that it is slight, is there any theoretical bound to it?

2. The paper briefly mentions possible unexplored interactions between MINTO and exploration strategies. Can the authors clarify how MINTO affects exploratory behavior?

---

> ### Author Response · Authors · 2025-11-26
> **Rebuttal (1/3)**
>
> We thank the reviewer for the detailed feedback and helpful suggestions. We have addressed the comments and revised the submission wherever changes were needed.
>
> > W1. The main evaluation is conducted against SimbaV1 and SimbaV2, which appear to be less strong baselines. This certainly raises the question whether this method can generalize to strong baselines such as Rainbow.
>
> **On the strength of the Simba baselines.** \
> SimbaV2 [1] is a very recently proposed algorithm (ICML 2025) and is considered one of the current state-of-the-art off-policy actor–critic methods, demonstrating substantial improvement over prior methods. In particular, Table 1 in [1] shows that SimbaV2 outperforms many recent baselines. We believe the confusion may stem from the specific experimental setting considered in our paper, where we study these methods in the single-estimator setting. The Simba methods typically employ a single estimator only for a subset of the task suites (e.g., DMC-Hard). In our work, we analyze the relative performance gain from integrating MINTO into such methods, both
>
> - in the single-estimator case (Fig. 7), and
> - in the two-estimator case, where we apply the Clipped Double Q-learning (CDQ) trick from TD3 [2] (Figs. 16–19 in the appendix).
>
> In the two-estimator setting, we integrate MINTO by including the online estimates in the minimum operator used in CDQ for computing the target.
>
> **Additional experiments on a different strong baseline.** \
> Motivated by the Reviewer’s suggestion, we also evaluated MINTO in the actor–critic setting on another strong baseline that is not part of the Simba family. Specifically, we integrated our method into CrossQ+WN [3], a very recent algorithm (accepted at NeurIPS 2025) that reintroduces the target network and shows significant improvements over its predecessor CrossQ [4], which removed the target network. We benchmarked CrossQ+WN and CrossQ+WN + MINTO on the three task suites considered in our work, Mujoco, HBench, and DMC-Hard, covering a total of 26 continuous-control tasks. The new results, reported in Figs. 22–24 in the appendix, consistently show performance gains thanks to integrating MINTO, both in the single-estimator and two-estimator (CDQ) settings. These additional experiments further support the generality and effectiveness of MINTO beyond the Simba family.
>
> **On Rainbow as a baseline.** \
> Regarding Rainbow [5], prior results reported in [6] show that IQN [7] generally provides stronger performance than Rainbow. Accordingly, we prioritized IQN over Rainbow for assessing the effectiveness of MINTO.

---

> ### Author Response · Authors · 2025-11-26
> **Rebuttal (2/3)**
>
> > W2. The exclusive reliance on the minimum operator can be too conservative, possibly causing underestimation and inhibiting exploration in low-noise or optimistic environments. \
> > Q1. Have you studied the impact of potential underestimation? While you claim that it is slight, is there any theoretical bound to it?
>
> **Rationale Behind the Use of the Minimum Operator in RL** \
> The use of a minimum operator has become a widely adopted technique for mitigating the overestimation bias introduced by the maximum operator in the Bellman optimality equation. This practice originated with TD3 [2], where the minimum over two value estimators (two critics) was proposed as a simple yet highly effective solution to overestimation, a technique now known as Clipped Double Q-Learning (CDQ). Since then, CDQ has been incorporated into numerous modern actor–critic algorithms, including SAC [8], SimbaV1 [9], SimbaV2 [1], and CrossQ+WN [3], where it remains a core design component that significantly contributes to performance (see Figs. 16–19 and Figs. 22 & 24). A similar idea was also introduced in the value-based setting via Maxmin DQN [10], which also uses a minimum operator to reduce overestimation.
>
> In contrast to these approaches, MINTO employs the minimum operator for a different purpose: to incorporate *fresh* online estimates into the target in a *stable* manner. Importantly, the online and target estimators in MINTO are not fully independent, they are correlated due to periodic hard updates (in DQN) or soft updates via Polyak averaging (in actor–critic methods). This correlation significantly reduces the risk of severe underestimation compared with methods such as Maxmin DQN or CDQ-based techniques, which rely on ideally independent estimators.
>
> **Car-on-the-Hill example supporting our claim** \
> To empirically study potential underestimation, in the revised version, we show a focused analysis on the Car-on-the-Hill environment from [11], comparing MINTO against DQN, Double DQN, and Maxmin DQN (see Fig. 33). This environment is a variant of Mountain Car for which it is possible to compute true optimal action values, as explained in [11]. We evaluated three metrics: average empirical bias, value estimation MSE, and average return.
> As expected, DQN shows clear overestimation bias, while Double DQN slighly reduces this effect. Maxmin DQN, which applies the minimum operator over two highly independent estimators, exhibits pronounced underestimation. In contrast, MINTO achieves substantially lower bias without the strong underestimation seen in Maxmin DQN. Moreover, the overshoot in bias observed in DQN and Double DQN does not appear in MINTO.
> Although MINTO’s estimation error is comparable to that of the baselines, we observe a small increase that yields a slightly higher variance, an expected consequence of incorporating a moving (online) component into the target. Nevertheless, MINTO solves the task *faster* than all other methods and does so without exhibiting learning *instability*, despite incorporating online estimates into the target. It is also worth noting that Car-on-the-Hill is a deterministic, low-noise environment, yet MINTO still performs reliably and does not suffer from the conservatism that would typically arise from severe underestimation. We believe that these results directly address the Reviewer’s concern.
>
> **Theoretical analysis as a future direction** \
> We agree that a theoretical analysis of the bias properties of MINTO, including deriving explicit bias bounds, would be a valuable complement to our empirical findings. We consider such analysis an important and natural direction for future work. We believe that the Car-on-the-Hill study provides strong empirical evidence that MINTO avoids significant underestimation while retaining stability and efficiency, and we look forward to extending our work with theoretical guarantees in future iterations.
>
> > W3. The code is not attached and will be released upon acceptance. This raises certain concerns.
>
> Thank you for pointing this out. We have attached the complete codebase for all three RL scenarios discussed in the paper. This should allow the Reviewers to fully inspect the implementation details and verify the reproducibility of our results.

---

> ### Author Response · Authors · 2025-11-26
> **Rebuttal (3/3)**
>
> > Q2. The paper briefly mentions possible unexplored interactions between MINTO and exploration strategies. Can the authors clarify how MINTO affects exploratory behavior?
>
> Despite the reasonable bias observed in the Car-on-the-Hill study, it is important to note that optimistic value estimates, arising from overestimation, can, in some cases, facilitate exploration. This may explain why, in a few games, DQN achieves better performance than MINTO. These observations motivate us to further investigate how the proposed operator influences exploratory behavior, as well as to explore the potential need for an adaptive operator that does not rely exclusively on the minimum operation.
>
> This point is related to (W2) raised by the Reviewer; however, our perspective differs slightly. Our motivation for examining the interaction between MINTO and exploration does not stem from any excessive conservativeness or severe underestimation caused by MINTO. Instead, it stems from the fact that MINTO maintains a reasonable estimation bias and is therefore less optimistic than methods that tend to overestimate value functions. In environments where such optimism is beneficial for exploration, this difference may matter. Consequently, studying how MINTO interacts with exploration strategies remains an interesting and valuable direction for future work.
>
>
> [1] Lee, Hojoon, et al. "Hyperspherical Normalization for Scalable Deep Reinforcement Learning." Forty-second International Conference on Machine Learning. \
> [2] Fujimoto, Scott, Herke Hoof, and David Meger. "Addressing function approximation error in actor-critic methods." International conference on machine learning. PMLR, 2018. \
> [3] Palenicek, Daniel, et al. "Scaling off-policy reinforcement learning with batch and weight normalization." arXiv preprint arXiv:2502.07523 (2025). \
> [4] Bhatt, Aditya, et al. "CrossQ: Batch Normalization in Deep Reinforcement Learning for Greater Sample Efficiency and Simplicity." The Twelfth International Conference on Learning Representations. \
> [5] Hessel, Matteo, et al. "Rainbow: Combining improvements in deep reinforcement learning." Proceedings of the AAAI conference on artificial intelligence. Vol. 32. No. 1. 2018. \
> [6] Agarwal, Rishabh, et al. "Deep reinforcement learning at the edge of the statistical precipice." Advances in neural information processing systems 34 (2021): 29304-29320. \
> [7] Dabney, Will, et al. "Implicit quantile networks for distributional reinforcement learning." International conference on machine learning. PMLR, 2018. \
> [8] Haarnoja, Tuomas, et al. "Soft actor-critic algorithms and applications." arXiv preprint arXiv:1812.05905 (2018). \
> [9] Lee, Hojoon, et al. "SimBa: Simplicity Bias for Scaling Up Parameters in Deep Reinforcement Learning." The Thirteenth International Conference on Learning Representations. \
> [10] Lan, Qingfeng, et al. "Maxmin Q-learning: Controlling the Estimation Bias of Q-learning." International Conference on Learning Representations. \
> [11] Ernst, Damien, Pierre Geurts, and Louis Wehenkel. "Tree-based batch mode reinforcement learning." Journal of Machine Learning Research 6 (2005).

---

> > ### Comment · Reviewer_t6g5 · 2025-11-26
> > **Thank you very much**
> >
> > I thank the author for addressing my concerns and performing more experiments. In particular, I found the results satisfactory and I appreciate the author's effort to enhance reproducibility. I have raised my score to 8.

---

> > > ### Author Response · Authors · 2025-11-26
> > >
> > > We would like to thank you for the score update. We are very pleased our rebuttal was effective in addressing your concerns.

---

### Official Review · Reviewer_Fczs · 2025-10-26

**Soundness:** 2
**Presentation:** 4
**Contribution:** 3
**Rating:** 4
**Confidence:** 4

**Summary:**

This paper proposes a new method for using the target network in RL.
Specifically, the authors modify the target computation by taking the minimum of the target Q-network and the online Q-network.
Albeit simple, it shows substantial improvements across several RL settings, including online RL (DQN, SAC (Simba-V1, Simba-V2)), distributional RL (IQN), and offline RL (CQL).

**Strengths:**

**S1. Simplicity of the method**
* The proposed method is conceptually simple and easy to implement; it only modifies the target calculation by taking the minimum between the target and online Q-values.
* This adjustment can be integrated easily into existing RL frameworks, as demonstrated in Algorithms 1–4.
Such simplicity makes the method practical and broadly applicable.

**S2. Substantial improvement across diverse RL domains**
* Despite its simplicity, the method demonstrates strong empirical performance.
* It improves results across various RL domains --- online, offline, and distributional settings --- and across both continuous and discrete action spaces.
* The reported gains over baseline methods in the DQN setting, suggest that the proposed approach effectively mitigates overestimation bias than other methods.

**S3. Clear organization and presentation**
* The paper is well structured and easy to read.
* Experimental results are clearly presented.

**Weaknesses:**

**W1. Comparison with other regularization methods limited in DQN setting**
* Currently, the comparison with other bias-reduction or regularization methods is limited to the DQN (discrete online RL) setting.
This restricts the strength of the empirical claims, especially since the method is positioned as a general improvement applicable across multiple RL domains.
* Including comparisons with well-established bias-reduction methods in other settings (e.g., SAC, IQN, and CQL) will clarify whether the observed improvements generalize beyond DQN.

**Questions:**

**Q1. Missing comparison with Clipped Q-learning and related baselines**
* To fully support the claim, the proposed method should be experimentally compared with other bias reduction methods (e.g., methods introduced in DQN experiment).
* Among those, I think at least **comparison with Clipped Q-learning [1] is necessary**, which is equivalent to MaxMin DQN with N=2 --- the strongest baseline reported for DQN in the paper, and also widely used.

I would like to emphasize that I believe this paper is strong and has substantial potential. However, this point is a critical concern for me. I currently lean toward a weak reject, but if the authors can provide the requested experimental comparisons (e.g., with Clipped Q-learning), I would be inclined to raise my recommendation, even significantly if the results meet the expectations.

---

> ### Author Response · Authors · 2025-11-26
> **Rebuttal (1/2)**
>
> We appreciate the reviewer’s thorough feedback and insightful suggestions, which we have incorporated into the updated version of the manuscript wherever necessary.
>
> > **W1. Comparison with other regularization methods limited in DQN setting** \
> > **Q1. Missing comparison with Clipped Q-learning and related baselines**
>
> We thank the Reviewer for raising this important point. In the revised version, we have substantially broadened the comparison beyond the DQN setting, including additional bias-reduction methods, where feasible. Below, we clarify the scope of our original experiments and describe the new results added during the rebuttal.
>
> > Currently, the comparison with other bias-reduction or regularization methods is limited to the DQN (discrete online RL) setting. This restricts the strength of the empirical claims, especially since the method is positioned as a general improvement applicable across multiple RL domains.
>
> **Why some comparisons were not included originally.**
>
> The main reason we did not include all desired combinations of baselines in the initial submission is the substantial computational cost associated with running them across our multiple RL settings, i.e., online RL, offline RL, distributional RL, and continuous control. Prior to the submission deadline, we did not have access to the resources necessary to run all these experiments thoroughly.
>
> During this rebuttal phase, we have been able to conduct a number of new experiments that directly address the Reviewer’s concerns. We believe that these new results significantly strengthen the empirical foundation of the paper, and we would like to thank the Reviewer again for encouraging us to do so.
>
> > Among those, I think at least comparison with Clipped Q-learning [1] is necessary, which is equivalent to MaxMin DQN with N=2 --- the strongest baseline reported for DQN in the paper, and also widely used.
>
> **Clarifying the role of Clipped Double Q-Learning (CDQ) -- Old Results**
>
> We fully agree with the Reviewer that including Clipped Double Q-Learning (CDQ), that was initially introduced in the TD3 paper [1], is an important part of the comparison. In fact, both SimbaV1 [2] and SimbaV2 [3], the two of the main actor-critic baselines in our work, are based on SAC [4], which already employs CDQ as a core component. We want to point out that **comparisons with CDQ were already present in the original submission** (see Figs. 16-19), but the legend labels used the term "Double Q", which may have caused confusion. We have corrected all relevant figures in the revised manuscript by replacing "Double Q" with "Clipped Double Q".
>
> Furthermore, given that the main objective of MINTO is to introduce recent online estimates into the target calculation, a feature absent from CDQ, we additionally evaluated MINTO when integrated directly into the CDQ update. In this variant, the online estimates are incorporated into the minimum operator used by CDQ, allowing us to assess the relative improvements that MINTO yields when combined with this widely used bias-reduction strategy.

---

> ### Author Response · Authors · 2025-11-26
> **Rebuttal (2/2)**
>
> > Including comparisons with well-established bias-reduction methods in other settings (e.g., SAC, IQN, and CQL) will clarify whether the observed improvements generalize beyond DQN.
>
> **Beyond the DQN setting.** \
> The Reviewer asked whether the baselines from Fig. 3 could be evaluated in settings beyond DQN. Although these baselines were originally proposed as DQN variants, we agree that examining them more broadly provides valuable insight into the generality of MINTO. We have conducted this analysis for this rebuttal, as explained in the following.
>
> **Adapting baselines to the Offline RL setting (CQL) -- New Results**
> Because CQL is closely related to DQN, adapting the DQN-based baselines to offline RL is doable. We implemented **Double CQL**, **FR-CQL**, **ScCQL**, and **Maxmin CQL** (N = 2), and evaluated all methods on the 14 Atari games, each run for 5 seeds following the standard offline RL protocol. These results are now presented in **Fig. 6 of the revised manuscript**.
>
> The results show that CQL+MINTO outperforms all single-estimator baselines, reflecting trends observed in the online setting. Interestingly, Maxmin CQL (N = 2) performs better than CQL+MINTO, which we attribute to the absence of exploration in offline learning, where conservative methods can be less detrimental but advantageous. Motivated by the benefits of introducing recent estimates into CDQ, we also integrated MINTO into Maxmin CQL, and again observed performance improvements. These findings reinforce the general benefit of the MINTO operator across different RL settings. For completeness, per-game learning curves are provided in Figs. 14–15 in the appendix.
>
> **Adapting baselines to the continuous control RL setting.** \
> As the Reviewer noted, Maxmin DQN is equivalent to Clipped Double Q-Learning (CDQ) in the actor–critic setting. These comparisons were already included in the original submission (see Appendix), and we clarified this earlier in our response. For the other baselines, the adaptation was less direct because they were originally introduced in the value-based literature. Among them, we found that Functional Regularization (FR-DQN) [5] extends most naturally to actor–critic methods, as it does not require redesigning the interactions between actor and critic components, unlike Double DQN [6] and ScDQN [7]. Yet, we argue that Double DQN is somewhat connected to the Clipped Double Q-Learning trick introduced in the TD3 [1] paper since both of them were inspired by the Double Q-Learning approach [8].
>
> Motivated by this reasoning, we introduced **FR-SimbaV1** and **FR-SimbaV2** by applying a functional regularization penalty that constrains the online critic relative to the target critic. We evaluated these methods on all 26 continuous-control tasks across the Mujoco, HBench, and DMC-Hard suites, running 10 seeds each, and considered both the single-estimator setting as well as the CDQ (double-estimator) setting for Mujoco and HBench, following the SimbaV2 protocol [3]. The results, shown in Figs. 25–30, indicate that SimbaV1+MINTO and SimbaV2+MINTO consistently outperform their FR-augmented counterparts across most tasks and estimator configurations.
>
> Additionally, following our discussions with Reviewer t6g5, we incorporated a new actor–critic baseline into our study: CrossQ+WN, a method recently accepted to NeurIPS 2025. We evaluated MINTO in combination with CrossQ+WN across all three task suites and observed consistent improvements, as reported in Figs. 22–24 (Sec. D.5.2). These findings align with our observations from the Simba family and further demonstrate the benefit of integrating MINTO into diverse actor–critic approaches.
>
> **Adapting baselines to the distributional RL setting.** \
> Integrating the DQN-based baselines into IQN is nontrivial, as it requires several additional design choices specific to the distributional setting. Running all such variants across the full set of 15 Atari games would also require substantially more computational resources than are feasible during the rebuttal period, especially given the longer training time of IQN compared to DQN. Moreover, we preferred to use the discussion period efficiently and avoid delaying our responses further.
>
> However, we note that SimbaV2 already employs distributional critics, and we have provided comparisons within that context, including CDQ and functional regularization.
>
> In addition, prior to the rebuttal period, we had already trained IQN and IQN+MINTO using the IMPALA architecture with LayerNorm to maintain consistency with our DQN and CQL experiments. These results are reported in Fig. 4 of the revised manuscript and show a clear performance gain resulting from the incorporation of recent online estimates via MINTO.

---

> ### Author Response · Authors · 2025-11-26
> **References**
>
> [1] Fujimoto, Scott, Herke Hoof, and David Meger. "Addressing function approximation error in actor-critic methods." International conference on machine learning. PMLR, 2018. \
> [2] Lee, Hojoon, et al. "SimBa: Simplicity Bias for Scaling Up Parameters in Deep Reinforcement Learning." The Thirteenth International Conference on Learning Representations. \
> [3] Lee, Hojoon, et al. "Hyperspherical Normalization for Scalable Deep Reinforcement Learning." Forty-second International Conference on Machine Learning.  \
> [4] Haarnoja, Tuomas, et al. "Soft actor-critic algorithms and applications." arXiv preprint arXiv:1812.05905 (2018).  \
> [5] Piché, Alexandre, et al. "Bridging the Gap Between Target Networks and Functional Regularization." Transactions on Machine Learning Research.  \
> [6] Van Hasselt, Hado, Arthur Guez, and David Silver. "Deep reinforcement learning with double q-learning." Proceedings of the AAAI conference on artificial intelligence. Vol. 30. No. 1. 2016.  \
> [7] Zhu, Rong, and Mattia Rigotti. "Self-correcting q-learning." Proceedings of the AAAI conference on artificial intelligence. Vol. 35. No. 12. 2021.  \
> [8] Hasselt, Hado. "Double Q-learning." Advances in neural information processing systems 23 (2010).

---

### Official Review · Reviewer_vd9n · 2025-10-31

**Soundness:** 4
**Presentation:** 3
**Contribution:** 3
**Rating:** 8
**Confidence:** 4

**Summary:**

The paper proposes a technique (MINTO) to make better use of the online network when also using a target network for learning the value function. The technique simply changes the update target value to be the minimum between the target network and online network's value estimates.
Experiments on a variety of benchmarks show that this technique is beneficial and can outperform common alternatives such as double Q-learning or using the minimum betweeen two target networks.

**Strengths:**

The paper was easy to follow and the proposed technique is very simple to understand and implement.
The experiments are comprehensive and show that method works in a variety of settings (Q-learning algorithms, actor-critic, distributional RL, continuous-control, atari).
I appreciated testing the variants that would be most relevant such as mean or max of the online and target network estimates.

In summary, I think the proposed technique would make a nice addition to the deep RL toolbox.

**Weaknesses:**

I did not identify any major weaknesses.

It could have been nice to include a more detailed analysis of why MINTO may be effective. In particular, thinking through the MINTO updates and comparing to using the target net as usual, the online network value replaces the target net value when the online network's value is lower. This means we can only use the most recent information (online network) if it is lowering the bootstrap target.
I wonder if you could study the accuracy of the value estimates in a policy evaluation setting where we could accurately estimate true value functions. Perhaps MINTO produces better estimates overall.

**Questions:**

Suggestions and comments(not directly impacting the score):
- In the ablations, it is reported that taking the mean between the online and target network estimates was not helpful. Do you have any hypotheses why this woudl be the case? Using the mean between two target critics has been found to sometimes an effective alternative to the min between two critics [1] and sometimes underestimation bias has been found to be harmful.


- The proposed method reminds me of the online network trust region method proposed in [2] since they both attempt to make use of the online network safely. It could be beneficial to include it as a baseline or provide some comparisons.

- The algorithm labelled  "Maxmin DQN" in Fig.3. seems to be very similar to the clipped Double-Q learning update proposed in the TD3 paper [3] and I had not seen any mention of this.


[1] "Scaling for compute and sample-efficient continous control" Nauman et al.

[2] "Human-level Atari 200x faster" Kapturowski et al.

[3] "Addressing function approximation error in actor-critic methods" Fujimoto et al.

---

> ### Author Response · Authors · 2025-11-26
> **Rebuttal (1/3)**
>
> We are grateful for the reviewer’s comprehensive comments and recommendations. These have been integrated into the manuscript, and we have made revisions wherever necessary.
>
> > W1. It could have been nice to include a more detailed analysis of why MINTO may be effective. In particular, thinking through the MINTO updates and comparing to using the target net as usual, the online network value replaces the target net value when the online network's value is lower. This means we can only use the most recent information (online network) if it is lowering the bootstrap target. I wonder if you could study the accuracy of the value estimates in a policy evaluation setting where we could accurately estimate true value functions. Perhaps MINTO produces better estimates overall.
>
> We thank the Reviewer for this insightful suggestion, which motivated us to analyze the accuracy of value estimates produced by MINTO, as well as the resulting estimation bias (see Sec. D4). To that end, we conducted a controlled study using the Car-on-the-Hill environment—a variant of Mountain Car specifically designed to allow accurate computation of the true optimal action-values $Q^*$, as discussed in [4]. This makes it an ideal setting for evaluating value-estimation accuracy under different learning algorithms. In our analysis, we benchmarked MINTO against DQN, Double DQN, and Maxmin DQN, all using the same simple neural network architecture to isolate the effect of the update rule. In Fig. 33 of the revised manuscript, we report three key metrics: average empirical bias, value-estimation MSE, and average return.
>
> As expected, DQN exhibits a clear overestimation bias, most noticeably through an overshoot in estimated values early in training. Double DQN reduces this effect slightly, but the overshoot remains noticeable. Maxmin DQN, which applies a minimum operator over two independent estimators, goes to the opposite extreme and produces systematic underestimation.
>
> In contrast, MINTO, by integrating recent online estimates through the minimum operator with the target network, reduces the overestimation observed in DQN and Double DQN, particularly eliminating the overshoot during early learning. At the same time, MINTO introduces significantly less underestimation than Maxmin DQN. In terms of value-estimation error, MINTO exhibits slightly higher variance, which is expected due to the introduction of a moving (online) component into the target. However, this variance does not hinder performance. Most importantly, when evaluating the actual task return, MINTO solves the Car-on-the-Hill problem the fastest and with the least fluctuation among all methods. This aligns directly with the goal of MINTO of achieving *fast* and *stable* reinforcement learning.

---

> ### Author Response · Authors · 2025-11-26
> **Rebuttal (2/3)**
>
> > Q1. In the ablations, it is reported that taking the mean between the online and target network estimates was not helpful. Do you have any hypotheses why this woudl be the case? Using the mean between two target critics has been found to sometimes an effective alternative to the min between two critics [1] and sometimes underestimation bias has been found to be harmful.
>
> We thank the Reviewer for this interesting insight. We agree that, in general, taking the mean across multiple critics can sometimes be beneficial, as shown in [1]. However, we believe there are important differences between our mean-based ablation (mean of online and target estimates) and the BRO method [1], which help explain why the mean variant was not effective in our setting.
>
> First, in our ablation, the mean operator is applied to **two correlated estimators**: the online and target networks, which are coupled through periodic hard updates. In contrast, BRO applies the mean operator over a set of **relatively independent estimators**. This distinction is analogous to the difference between our use of the minimum operator and methods such as Maxmin DQN [5] or the Clipped Double Q-Learning (CDQ) trick, where the critics are designed to be more independent.
>
> Second, as discussed in the BRO paper, the *architectural regularization* in BRO plays a crucial role in alleviating overestimation, thereby reducing the need for aggressive bias-reduction schemes such as CDQ. BRO also relies on a *distributional RL* formulation, and distributional methods are known to be relatively less affected by overestimation. This combination of architectural regularization and distributional learning likely makes the mean operator more suitable in BRO’s setting than in the standard value-based/deep Q-learning setups we consider.
>
> This perspective is consistent with observations from SimbaV2, which also enforces strong regularization at the architectural level and trains its critics using distributional RL. SimbaV2 is able to report surprisingly strong performance even without the CDQ trick and with only a single critic. Moreover, SimbaV2 is considered a stronger off-policy actor–critic method than BRO [1] (see Table 1 in the SimbaV2 paper [6]).
>
> In summary, our hypothesis is that in our setting, the mean of **correlated** online–target estimates does not sufficiently reduce overestimation to compensate for the additional noise induced by the moving online network. In contrast, in methods like BRO where multiple critics are more independent, strong architectural regularization is applied, and distributional learning mitigates overestimation, the mean operator and its optimistic nature become more beneficial in the absence of a pronounced overestimation problem.
>
> > Q2. The proposed method reminds me of the online network trust region method proposed in [2] since they both attempt to make use of the online network safely. It could be beneficial to include it as a baseline or provide some comparisons.
>
> We appreciate the Reviewer for highlighting this connection and for pointing us to the relevant work. Indeed, the approach proposed in [2], known as MEME, shares a conceptual motivation with MINTO in the sense that both attempt to leverage recent online estimates in a safe and stable manner. However, MEME is a comprehensive algorithm that combines several distinct components, each contributing to the final performance. We believe the Reviewer is specifically referring to the trust-region component (A1 in [2]), which relies on two stability conditions (Eqs. 1 and 2 in [2]).
>
> Motivated by the Reviewer’s suggestion, we attempted to implement this trust-region mechanism as a baseline. In doing so, we encountered a technical issue: enforcing Condition 2 (Eq. 2 in [2]) requires an estimate of the return, which is not accessible or computable within our implementation setup. As a result, we implemented only Condition 1 (Eq. 1 in [2]) and additionally incorporated the TD-error normalization proposed in Component B1. Unfortunately, this partial implementation proved to be highly unstable in practice, and we were unable to obtain meaningful or reliable results.
>
> To avoid presenting results that may be misleading, either due to implementing only a subset of MEME’s components or due to limitations of our own implementation, we decided **not to include this baseline in our empirical comparisons**. Instead, we have added a discussion of this related work in the revised manuscript to acknowledge its conceptual relevance. For transparency, we have also included our implementation attempt in the released code (see `trdqn.py` in the minto_onlinerl folder).

---

> ### Author Response · Authors · 2025-11-26
> **Rebuttal (3/3)**
>
> > Q3. The algorithm labelled "Maxmin DQN" in Fig.3. seems to be very similar to the clipped Double-Q learning update proposed in the TD3 paper [3] and I had not seen any mention of this.
>
> We thank the Reviewer for pointing out this connection. Indeed, the algorithm labeled "Maxmin DQN" in Fig. 3 is closely related to the Clipped Double Q-learning (CDQ) update introduced in the TD3 paper [3] within the actor–critic literature. In the original submission, comparisons with CDQ were included in the appendix (Sec. D.5) under the legend label "Double Q", but we acknowledge that this naming was unclear and caused confusion. In the revised manuscript, we have replaced the ambiguous "Double Q" label with "Clipped Double Q" to make this explicit.
>
> We also recognize that we did not explicitly cite TD3 in this context. This was an oversight on our part: our discussion focused on SAC, the underlying algorithm for both SimbaV1 and SimbaV2, which itself incorporates the CDQ update from TD3. We have now added the appropriate references to TD3 to clarify this lineage.
>
> In the main paper, our initial focus was on the single-estimator versions (No CDQ) of SimbaV1 and SimbaV2, motivated by observations in the SimbaV2 paper that single-critic variants can sometimes perform surprisingly well. This allowed us to highlight the benefit of MINTO in a simplified setting that resembles a similar target computation as in DQN, CQL, and IQN.
>
> The central objective of MINTO is to introduce fresh online estimates to facilitate faster and more stable learning. For this reason, we also investigated integrating MINTO into actor–critic methods while using the CDQ trick. Specifically, we introduced the online estimates into the minimum operator of CDQ. The results of this integration were already included in the appendix of the original submission under the legend “Double Q” (now revised to “Clipped Double Q”).
>
> Furthermore, in light of the rebuttal discussions with the other Reviewers, we have added additional results in the actor–critic setting. Specifically, we extended our comparisons to include the integration of Functional Regularization (adapted from FR-DQN) [7] into SimbaV1 and SimbaV2; these results are now presented in the appendix of the revised manuscript (see Sec. D.5.3). We also incorporated experiments involving CrossQ+WN [8], a recently accepted off-policy actor–critic method (NeurIPS 2025), reported in Sec. D.5.2. This allowed us to study the effect of MINTO on a more diverse class of actor–critic algorithms beyond the Simba family. In these new experiments, we additionally examined the impact of integrating MINTO into target updates that employ the CDQ trick, further illustrating how MINTO interacts with widely used bias-reduction mechanisms in modern actor–critic methods.
>
>
> [1] "Scaling for compute and sample-efficient continous control" Nauman et al.
>
> [2] "Human-level Atari 200x faster" Kapturowski et al.
>
> [3] Addressing function approximation error in actor-critic methods" Fujimoto et al
>
> [4] Ernst, Damien, Pierre Geurts, and Louis Wehenkel. "Tree-based batch mode reinforcement learning." Journal of Machine Learning Research 6 (2005).
>
> [5] Lan, Qingfeng, et al. "Maxmin Q-learning: Controlling the Estimation Bias of Q-learning." International Conference on Learning Representations.
>
> [6] Lee, Hojoon, et al. "Hyperspherical Normalization for Scalable Deep Reinforcement Learning." Forty-second International Conference on Machine Learning.
>
> [7] Piché, Alexandre, et al. "Bridging the Gap Between Target Networks and Functional Regularization." Transactions on Machine Learning Research.
>
> [8] Palenicek, Daniel, et al. "Scaling off-policy reinforcement learning with batch and weight normalization." arXiv preprint arXiv:2502.07523 (2025).

---

### Official Review · Reviewer_HyDL · 2025-11-02

**Soundness:** 3
**Presentation:** 3
**Contribution:** 3
**Rating:** 6
**Confidence:** 3

**Summary:**

for the bootstrapping target in deep Q-learning, uses the minimum of the online net and target net value estimates, instead of using just the target net value estimate

they call their algorithm "MINTO"

**Strengths:**

applies MINTO on many base algorithms (DQN, IQN, CQL, SAC) and architectures (CNN, IMPALA, SimbaV1, SimbaV2), and fairly consistently shows some improvements in returns

no additional hyperparameters

simple

a good number of fair baseline algorithms -- for ex, includes mean, max, random, and min of {online, target} nets in Fig 1, and later also includes Double DQN, FR-DQN, ScDQN, and Maxmin DQN

novel as far as I am aware

written clearly

**Weaknesses:**

I think it would be great if the paper were even more clear about how the hyperparameters were set. while Appendix C.3 says "All methods are run with their default hyperparameters", it's not immediately clear to me how the values in Table 1 were set. are they the default hyperparameters from any existing codebase? if not, why not, and how were they set?

5 seeds is not many seeds. (but the experiments cover so many base algorithms, architectures, codebases, and environment suites that I suspect this is not a big issue)

no tests on toy tasks, along the lines of Baird's counterexample. I think there is a non-negligible chance that MINTO will get "stuck" on some such problems, where both TD and DQN sometimes empirically require their loss to increase before it will decrease. see for ex Fig 1e here: https://openreview.net/pdf?id=j3bKnEidtT. MINTO empirically appears to work on a wide variety of other benchmarks though, so this whole bullet might be irrelevant (even including the possible case that MINTO fails on some toy tasks).

a bit verbose, for ex the paper calls MINTO "simple yet effective" 4 times (once with a comma after simple)

**Questions:**

"$Q$-learning offers a recursive update to approximate the state-action value function $Q$"

Should that last $Q$ be $Q^*$ instead of $Q$?

&nbsp;

does the "preliminaries" section allow for stochastic rewards? if not, is that intentional?

&nbsp;

"The foundation of this success lies primarily in Deep RL, initiated by the introduction of the Deep Q-Network (DQN) (Mnih et al., 2013), which marked the first successful application of deep neural networks in RL."

Neural Fitted Q Iteration was also a success, even though DQN could maybe be considered an even bigger success

&nbsp;

"A problem that presents a challenge and obstacle"

why both?

&nbsp;

"In deep RL, the problem of moving targets is especially evident due to the use of neural networks and the resulting uncontrolled fluctuations in the values of unseen states."

what is this contrasting against? linear RL?

&nbsp;

"For example, Gallici et al. (2025) demonstrated that cleverly using parallel environments eliminates the need for a target network"

some people might consider "eliminates the need for a target network" as too strong of a claim

&nbsp;

"This makes MellowMax orthogonal to our approach."

you might say "somewhat orthogonal" to help avoid confusion

&nbsp;

"making this method orthogonal to our approach"

likewise

&nbsp;

"known as $Q$-function"

typo, should be "known as _a_ $Q$-function" (or "_the_ $Q$-function")

&nbsp;

"(Mnih et al., 2013) introduce a series of algorithmic components, more importantly, is the introduction of the target networks."

typos

&nbsp;

"Despite the success, this results in a slow learning of value function as well as the policy due to relying on out-dated estimates"

typo, should be "learning of _the_ value function"

&nbsp;

"bellman"

typo, should be "Bellman"

&nbsp;

"can we find a practical bellman update rule that results in a stable and fast learning?"

what does "practical" specify here?

&nbsp;

"When implemented in an efficient deep learning framework such as JAX (Bradbury et al., 2018), this overhead is negligible."

I don't understand. does JAX automatically parallelize the additional forward pass with the target network's forward pass?

&nbsp;

"addressing Q1.As"

typo

&nbsp;

why is Fig 1 with 10 Atari games, but Fig 2 with 15 games?

&nbsp;

Fig 2 shows CNN and IMPALA results on 100 vs 50 million frames. why not both 100 or both 50?

&nbsp;

"into both value-based and actor–critic methods"

this is maybe slightly confusing wording because actor-critic methods are partially value-based. maybe "both actor-critic and purely value-based methods" would be slightly clearer.

&nbsp;

"Offline RL aims to learn an optimal policy from a large and static dataset"

offline RL does not need a "large" dataset to be offline RL

&nbsp;

"A central challenge in this paradigm is the distributional shift problem: the learned policy may query the Q-function on state–action pairs absent from the dataset"

offline policy evaluation does not necessarily involve a learned policy. offline policy evaluation is also offline RL, and also may query the value function on state or state-action inputs not present in the dataset

---

> ### Author Response · Authors · 2025-11-26
> **Rebuttal (1/3)**
>
> We thank the reviewer for the extensive feedback and the many valuable suggestions. We have incorporated the feedback into the revised version of the manuscript, including correcting all identified typos and adopting the recommended improvements to the writing.
>
> > W1. I think it would be great if the paper were even more clear about how the hyperparameters were set. while Appendix C.3 says "All methods are run with their default hyperparameters", it's not immediately clear to me how the values in Table 1 were set. are they the default hyperparameters from any existing codebase? if not, why not, and how were they set?
>
> We thank the Reviewer for highlighting this point. We agree that our original description of how hyperparameters were selected was not sufficiently detailed, and we have improved this in the revised manuscript.
>
> **Atari experiments.**
>
> For the Atari suite, we briefly stated that our experiments follow the evaluation protocol of [1] and referenced [2] for implementation guidelines when benchmarking IQN. However, we did not explicitly clarify that **all hyperparameters for DQN and IQN in the online RL setting, as well as for CQL in the offline RL setting, were taken directly from the Dopamine framework [2]**. For the baselines shown in Fig. 3, since they are all DQN variants, we used the same hyperparameters as DQN unless otherwise specified in the original papers. Task-specific hyperparameters were adopted from the recommended values in the corresponding publications.
>
> **Continuous-control experiments.**
>
> For the continuous-control tasks, **all hyperparameters were taken from the SimbaV2 paper** [3]. As mentioned in the appendix, our implementation of these methods is based on the official SimbaV2 codebase [4], and therefore inherits its default settings.
>
> To improve transparency and address the Reviewer’s concern, we have added explicit sentences in the revised manuscript (see Sec. 5.1 & 5.3) clarifying the source of all hyperparameters. Additionally, we have included the codebases used for our experiments, along with configuration and run files, to ensure full reproducibility.
>
> > W2. 5 seeds is not many seeds. (but the experiments cover so many base algorithms, architectures, codebases, and environment suites that I suspect this is not a big issue)
>
> We thank the Reviewer for the comment. As noted in the previous point, our experimental setup for the Atari suite follows the Dopamine framework [2], which adopts 5 seeds as the default protocol. We chose to follow the same convention to ensure consistency with prior work and comparability with standard baselines. Moreover, as discussed in [5], using 5 seeds is generally considered sufficient for evaluating RL methods in the Atari suite.
>
> For the continuous-control experiments, we again followed common practice in the literature. In particular, SimbaV2, on which our implementation is based, uses 10 seeds, and we adopted the same number for all continuous-control results to maintain consistency and reliability.
>
> > W3. no tests on toy tasks, along the lines of Baird's counterexample. I think there is a non-negligible chance that MINTO will get "stuck" on some such problems, where both TD and DQN sometimes empirically require their loss to increase before it will decrease. see for ex Fig 1e here: https://openreview.net/pdf?id=j3bKnEidtT. MINTO empirically appears to work on a wide variety of other benchmarks though, so this whole bullet might be irrelevant (even including the possible case that MINTO fails on some toy tasks).
>
> We thank the Reviewer for the valuable comment and for pointing us to the relevant reference. Following the suggestion, we conducted an additional study on the Car-on-the-Hill environment to analyze performance and training loss, as well as the estimation error and bias exhibited by several methods, including DQN, Double DQN, Maxmin DQN, and MINTO (see Sec. D4 in the revised manuscript). In Fig. 33, we report the training loss alongside the estimation error and performance metrics.
>
> Interestingly, the behavior we observe in the training loss mirrors the phenomenon described in the referenced work. Specifically, all methods, including MINTO, show an initial increase followed by a decrease in training loss. However, the magnitude of this fluctuation is notably smaller for MINTO. We have added this discussion to Sec. D4 of the revised manuscript and included a citation to the suggested work.
>
> Importantly, MINTO successfully solved the Car-on-the-Hill task the fastest, without fluctuations in performance and without getting stuck in suboptimal solutions.

---

> ### Author Response · Authors · 2025-11-26
> **Rebuttal (2/3)**
>
> > W4. a bit verbose, for ex the paper calls MINTO "simple yet effective" 4 times (once with a comma after simple)
>
> We appreciate the Reviewer’s attention to detail and the suggestion regarding verbosity. In response, we have revised the manuscript to reduce repetition, particularly regarding the phrase "simple yet effective", and have generally streamlined the wording to improve clarity and readability.
>
> > Q1. does the "preliminaries" section allow for stochastic rewards? if not, is that intentional?
>
> We thank the Reviewer for pointing that out. Yes, our preliminaries should have considered the stochastic case. That said, we have modified this part in our revised version of our submission.
>
> > Q2. "The foundation of this success lies primarily in Deep RL, initiated by the introduction of the Deep Q-Network (DQN) (Mnih et al., 2013), which marked the first successful application of deep neural networks in RL."
> > Neural Fitted Q Iteration was also a success, even though DQN could maybe be considered an even bigger success
>
> Thank you for the suggestion. We have revised the sentence to properly acknowledge Neural Fitted Q Iteration as an important early work and as part of the foundations of Deep RL.
>
> > Q3. "A problem that presents a challenge and obstacle" why both?
>
> Thank you for pointing this out. The phrase was redundant, so we have revised it for clarity and now use only "an obstacle" in the updated manuscript.
>
> >  Q4. "In deep RL, the problem of moving targets is especially evident due to the use of neural networks and the resulting uncontrolled fluctuations in the values of unseen states." what is this contrasting against? linear RL?
>
> This statement is intended to contrast Deep RL with the tabular RL setting. In tabular methods, value estimates are stored explicitly for each state or state–action pair, which avoids the generalization effects that arise with neural network function approximation. In Deep RL, by contrast, the use of neural networks can produce uncontrolled fluctuations in the estimated values of unseen or rarely seen states, making the moving-target problem more pronounced.
>
> > Q5. "This makes MellowMax orthogonal to our approach." you might say "somewhat orthogonal" to help avoid confusion
>
> Thank you for the suggestion. To avoid the impression of an absolute statement, we have revised the wording for clarity. In the updated manuscript, we state that MellowMax is "mostly" orthogonal to our approach.
>
> > Q6. "making this method orthogonal to our approach" -> likewise
>
> Thank you for the comment. Our intention was to convey that the follow-up work adopts a different strategy for stabilizing value targets, rather than to imply that the approaches are entirely unrelated. In fact, this distinction enables our proposed method, MINTO, to be combined with CrossQ+WN [1], as shown in the revised manuscript. We now include this combination as an additional empirical result (alongside the Simba family), further demonstrating the effectiveness of MINTO in improving the performance of various actor–critic algorithms.
>
> > Q7. "can we find a practical bellman update rule that results in a stable and fast learning?" what does "practical" specify here?
>
> By "practical", we refer to a Bellman update rule that remains effective and robust in Deep RL settings, where value (Q) functions are approximated with neural networks. In other words, we are interested in update formulations that are not only theoretically sound but also feasible to implement and stable in large-scale function-approximation regimes.
>
> > Q8. "When implemented in an efficient deep learning framework such as JAX (Bradbury et al., 2018), this overhead is negligible." I don't understand. does JAX automatically parallelize the additional forward pass with the target network's forward pass?
>
> We stated this for two reasons. First, JAX is generally relatively faster than other deep learning frameworks due to its compilation (e.g., via XLA), so the additional forward pass adds little overhead. Second, JAX’s asynchronous dispatch mechanism allows independent computations to be enqueued without blocking, which often reduces the effective cost of extra operations. This behavior is described in the JAX async-dispatch documentation [6] and discussed in the linked thread [7]. We note, however, that asynchronous dispatch does not guarantee full parallel execution in all cases. While we could explicitly parallelize the two forward passes using vmap, in our implementation, we did not observe a need for further optimization.

---

> ### Author Response · Authors · 2025-11-26
> **Rebuttal (3/3)**
>
> > Q9. why is Fig 1 with 10 Atari games, but Fig 2 with 15 games?
>
> Thank you for pointing this out. The main evaluation of the online RL methods was conducted on all 15 Atari games. However, for the ablation study in Fig. 1, we were not able to complete experiments on all 15 games before the submission deadline due to limited computational resources. Therefore, we included results for only 10 games. In the revised manuscript, we have completed the experiments for the remaining games and updated Fig. 1 accordingly (as well as the corresponding figures in the Appendix).
>
> > Q10. Fig 2 shows CNN and IMPALA results on 100 vs 50 million frames. why not both 100 or both 50?
>
> Thank you for the question. The IMPALA-based methods require substantially more computation per environment step, resulting in training times that are nearly twice as long as those of the CNN-based methods. Running IMPALA for 100 million frames would have exceeded the 24-hour job time limit on our available cluster. For this reason, we trained IMPALA-based methods for 50 million frames, while retaining the 100 million frames for CNN-based methods for completeness. Importantly, for fairness, all methods using the same architecture (CNN or IMPALA) are compared under the same number of training frames.
>
> > Q11. "A central challenge in this paradigm is the distributional shift problem: the learned policy may query the Q-function on state–action pairs absent from the dataset" -> offline policy evaluation does not necessarily involve a learned policy. offline policy evaluation is also offline RL, and also may query the value function on state or state-action inputs not present in the dataset
>
> We appreciate the Reviewer’s clarification. In this paragraph, our intention was to describe the challenge in the context of Deep RL settings where neural networks are used to approximate value (Q) functions. In such cases, distributional shift typically arises because the value function (or Q-function) may be queried on state or state–action pairs that are not well represented in the dataset, regardless of whether the policy being evaluated is learned or fixed. We agree with the Reviewer that offline policy evaluation is a component of offline RL and can also suffer from this issue. Therefore, our discussion is not intended to be limited to learned policies, but rather to highlight the general relevance of distributional shift in value-function approximation.
>
> [1] Machado, Marlos C., et al. "Revisiting the arcade learning environment: Evaluation protocols and open problems for general agents." Journal of Artificial Intelligence Research 61 (2018): 523-562.
>
> [2] Castro, Pablo Samuel, et al. "Dopamine: A research framework for deep reinforcement learning." arXiv preprint arXiv:1812.06110 (2018).
>
> [3] Lee, Hojoon, et al. "Hyperspherical Normalization for Scalable Deep Reinforcement Learning." Forty-second International Conference on Machine Learning.
>
> [4] https://github.com/DAVIAN-Robotics/SimbaV2/tree/master
>
> [5] Agarwal, Rishabh, et al. "Deep reinforcement learning at the edge of the statistical precipice." Advances in neural information processing systems 34 (2021): 29304-29320.
>
> [6 https://docs.jax.dev/en/latest/async_dispatch.html
>
> [7] https://github.com/jax-ml/jax/discussions/8281

---

### Author Response · Authors · 2025-11-26
**General Comment**

We thank all Reviewers for their thoughtful and constructive feedback. In our rebuttal, we aim to address all concerns comprehensively and clear any points of confusion. We want to clarify that the delay in our response is due to the time we needed to complete the requested experiments to provide a thorough, unified reply to all Reviewers.

As part of the rebuttal process, we have updated the paper by incorporating revisions based on the reviewers’ comments. Newly added text is highlighted in blue for clarity. Additionally, we have included the codebase used to run the experiments, as requested by one of the Reviewers. We summarized the main additions included in our rebuttal, in the light of the reviews:

- **Adaptation of the baselines in Fig. 3 to the offline RL setting** and reporting results for Double CQL, FR-CQL, ScCQL, and Maxmin CQL on 14 Atari games, each run with 5 seeds (see Fig. 6 in the main paper and Figs. 14 & 16 in the Appendix) &#8594; Asked by **Reviewer Fczs**.
- **Adaptation of the Functional Regularization (FR-DQN) baseline to continuous control** by integrating it into SimbaV1 and SimbaV2 and evaluating it on all 26 continuous-control tasks, each run with 10 seeds (see Figs. 25–30 in Sec. D.5.3) &#8594; Asked by **Reviewer Fczs**.
- Correction of the legend label "Double Q" to **"Clipped Double Q" for the additional experiments requested by the Reviewer**, experiments that were already included in the original submission but were inaccurately labeled (see Figs. 16–19 in Sec. D.5.1) &#8594; Asked by **Reviewers Fczs & vd9n**.
- Inclusion of a recently accepted off-policy actor–critic method, CrossQ+WN, as **an additional strong baseline to further evaluate the generality of MINTO** across a diverse set of algorithms beyond the Simba family (see Figs. 22–24 in Sec. D.5.2) &#8594; Asked by **Reviewer t6g5**.
- Addition of the Car-on-the-Hill experiment to **analyze the value-estimation error and bias of MINTO and related baselines**, as well as to **investigate the observed increase-then-decrease pattern in training loss and whether MINTO may get stuck** on such toy examples (see Sec. D.4) &#8594; Asked by **Reviewers HyDL & vd9n & t6g5**.

---

### Meta-Review · Area_Chair_ikz1 · 2026-01-06

**Summary:**

MINTO proposes a minimal change to bootstrapping in deep RL: compute the TD target using the minimum of the online network and the target network value estimates, with the goal of injecting “fresh” estimates without losing target-network stability. The method is attractive because it introduces no new hyperparameters, is easy to integrate into many algorithms, and is evaluated broadly (online DQN/IQN, offline CQL, and continuous-control actor–critic variants including Simba and CrossQ+WN), where it generally improves sample efficiency and final returns.

**Reviewer Concerns:**

During rebuttal, the authors addressed the main substantive weaknesses: they clarified hyperparameter sources (Dopamine for Atari and CQL; SimbaV2 defaults for continuous control) and released code/configs; added toy-task analysis (Car-on-the-Hill) showing MINTO reduces early overshoot/overestimation and solves the task faster without getting stuck; corrected confusion around “Double Q” vs “Clipped Double Q” and expanded comparisons to clipped double-Q, functional regularization baselines in continuous control, and additional offline/CQL adaptations of strong DQN-regularizers. They also discussed potential underestimation/exploration trade-offs candidly, and expanded empirical coverage to address prior requests; importantly, a key reviewer explicitly raised their score to accept after these additions.

**Reviewer Scores:**

The review set is now clearly positive: two strong accepts (8/10) and one clear upgrade from borderline (4→8) after rebuttal experiments and code release. The remaining concerns are largely “nice-to-have” (deeper theory of bias bounds, more detailed analysis of exploration interactions), and do not appear to block acceptance given the simplicity and breadth of validated improvements.

---

### Decision · Program_Chairs · 2026-01-26

Accept (Poster)